# Genetic controllers for enhancing the evolutionary longevity of synthetic gene circuits in bacteria

Daniel P. Byrom & Alexander P. S. Darlington ✉

Engineered gene circuits often degrade due to mutation and selection, limiting their long-term utility. Here we present designs for genetic controllers which maintain synthetic gene expression over time. Using a multi-scale "host-aware" computational framework, which captures interactions between host and circuit expression, mutation, and mutant competition, we evaluate several controller architectures based on three metrics for evolutionary stability: total protein output, duration of stable output, and half-life of production. We propose a number of designs with varying inputs (e.g., output per cell, growth rate) and actuation methods (transcriptional vs. post-transcriptional regulation). We find post-transcriptional controllers generally outperform transcriptional ones, but no single design optimizes all goals. Negative autoregulation prolongs short term performance, while growth-based feedback extends functional half-life. We propose three biologically feasible, multi-input controllers that improve circuit half-life over threefold without requiring coupling the process to an essential gene or a genetic kill switch.

Synthetic biology seeks to engineer living organisms, often microbes, to perform desired functions, with a range of applications in healthcare, the chemicals industry and environmental science[1–3]. However, loss or degradation of function over microbial generations shortens the lifespan of such systems and represents a fundamental roadblock to widespread adoption and application in industry[4,5].

Engineered synthetic gene networks, often referred to as "circuits", utilise their host's gene expression resources, such as ribosomes and amino acids, for their own expression. This disrupts the cell's natural homeostasis as these resources are diverted away from host processes and towards circuit gene expression. This additional load imparted by the circuit often leads to a gross reduction in the cell growth rate, a phenomenon known as "burden" (reviewed extensively in refs. 6–9). In microbes, such as *E. coli*, where growth rate is analogous to fitness, cells which contain gene circuits are at a selective disadvantage as they cannot produce new daughter cells as quickly as their faster-growing, unengineered counterparts.

DNA replication is an inherently error-prone process, so every cell division represents a possibility for such mutations to be introduced into a gene circuit. It is therefore inevitable that they will eventually arise within a large population[10]. Mutations to promoters, ribosome binding sites or transcription factor binding sites can result in significant changes to gene circuit dynamics. Where these mutations inhibit circuit function and correspondingly reduce cellular resource consumption, the aforementioned growth disparity enables the new mutant strains to outcompete the ancestral strain. As a result, synthetic gene circuit function is eventually eliminated from engineered populations[4] (Fig. 1). In some cases, such evolutionary loss-of-function can occur so rapidly that a culture cannot be grown to a suitable size before its effects become significant[11].

The evolutionary longevity of a gene circuit can be quantified by measuring the time taken for the population-level output to reach a predefined "breaking point"[12] (e.g., for a simple output-producing circuit, the "half-life" describes the time taken for the output to fall by 50%). A number of experimental approaches have been developed which aim to improve the evolutionary longevity of gene circuits (reviewed extensively in refs. 13,14). These are divided into two broad approaches: (1) suppressing the emergence of circuit mutants and (2) reducing their selective advantage[15]. Examples of approach (1) include engineering host organisms with reduced mutation rate[16] and reducing the use of

Warwick Integrative Synthetic Biology Centre, School of Engineering, University of Warwick, Coventry, UK. ✉e-mail: a.darlington.1@warwick.ac.uk

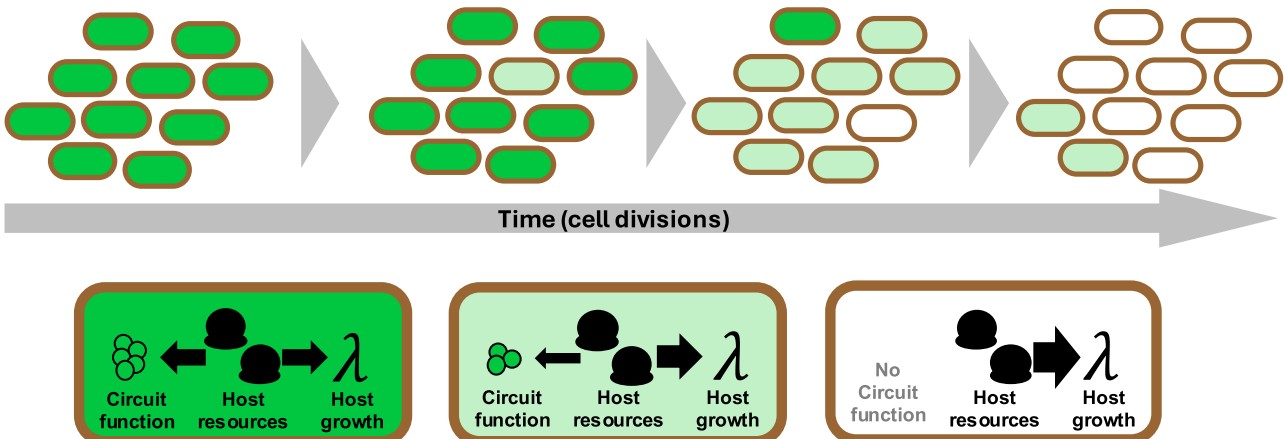

**Time (cell divisions)**

**Fig. 1 | A schematic demonstrating how function is lost in an engineered population over evolutionary time.** Random mutations occur as a result of error-prone DNA replication, leading to mutants with reduced circuit function. These mutants have higher selective fitness, as the host cells can direct more resources towards growth. As a result, such mutants dominate the population and circuit function is eventually lost.

repeated DNA sequences[4,10]. Examples of approach (2) include systems where circuit function is artificially coupled to host survival[17–19]. Such methods are often bespoke and may constrain the performance/function of a synthetic device. For example, in ref. 18, Yang et al. engineered a bidirectional promoter that simultaneously drives both GFP expression and antibiotic resistance with the view that mutations to the shared promoter ought to be selectively disadvantageous as antibiotic resistance is lost. Although the promoter sequence is maintained after serial passaging, the system remains prone to mutation in the ribosome binding site. In general, even when methods are implemented to extend the longevity of gene circuits, evolution is never entirely mitigated[14] and some designs still lose expression within 24 h[20].

Maintenance of function in light of evolution is still an open challenge for engineering biology, with a need for design frameworks to predict and account for the mutation during the design process[5,21,22]. Here, we tackle the problem from a systems perspective, aiming to identify feedback controller design paradigms that can improve evolutionary longevity. In this paper, we view the onset of mutation as a parametric uncertainty and the competition between mutant and ancestral strains as an environmental perturbation. Negative feedback is an attractive control strategy for minimising the impact of both perturbations and uncertainty because it enables a system to monitor its output and feed this information back into the system as an input, thereby adjusting its behaviour to maintain a known level. Negative feedback has been successfully implemented in synthetic biology for a variety of objectives[23,24], including optimising yields in metabolic engineering[25] and reducing the unexpected coupling between genetic parts (e.g.[26]). It has been shown both theoretically and experimentally to reduce burden and improve the evolutionary longevity of synthetic circuits[27–30]. However, existing work typically demonstrates that this reduction in burden is at least partly a result of reduced expression of the target genes (i.e., reduced circuit function), so the closed-loop systems are not compared against open-loop systems of equivalent function. Further, to date, little attention has been paid to how negative feedback controllers should be optimally designed to improve evolutionary longevity, both in terms of controller topology and design rules for part selection. By taking a systems engineering approach and developing mathematical models, we are able to investigate these considerations in silico.

Here, we use a host-aware design framework to analyse different design choices for the creation of genetic controllers which enhance the long-term performance of a simple genetic circuit. This model accounts for host-circuit interactions, dynamic growth and mutation, enabling us to evaluate the relationships between circuit expression, burden and evolutionary population dynamics[30–34]. To evaluate evolutionary longevity, we consider both the maintenance of function in the short term (by measuring how long function is maintained within a narrow window around the designed level) and the persistence of the circuit in the long term (by measuring how long it takes for function to halve). We demonstrate the performance of a variety of controller architectures which differ in the quantity sensed by the system as the 'control input', and the mechanism through which control is enacted. In each case, we evaluate how to optimally design the controllers, the potential impact of additional controller resource consumption and their robustness to parametric variation to guide in vivo implementation.

We demonstrate three key factors which determine the effectiveness of genetic controllers for enhancing evolutionary longevity. Firstly, the choice of controller input: growth-based feedback significantly outperforms intra-circuit feedback and population-based feedback in the long term, while intra-circuit feedback can provide significant improvements in the short term. Secondly, the means of controller actuation: we show that post-transcriptional control, which exploits small RNAs (sRNA) to silence circuit RNA, outperforms transcriptional control via transcription factors, as this mechanism provides an amplification step which enables strong control with reduced controller burden. Thirdly, we show that systems with separate circuit and controller genes can exhibit significantly enhanced performance due to evolutionary trajectories where loss of controller function results in short-term increases in protein production. We also show that different controller architectures have different robustness to parametric uncertainty, which will complicate design selection for in vivo implementation. We propose a controller topology that combines control inputs and feedback mechanisms to improve both short- and long-term performance while maintaining enhanced robustness to parametric uncertainty.

## Results
### Modelling and quantifying gene circuit evolution
We develop an ordinary differential equation model of host-circuit interactions[31,33] and augment it with a model describing an evolving population of E.coli cells, taking a similar state-transition approach to ref. 34 and ref. 30. This multi-scale model comprises a set of competing populations sharing a single source of nutrients, with each population representing a different parameterisation (or "strain") of an engineered E. coli cell. Mutation is implemented via transitions between these different strains, and selection emerges dynamically through differences in calculated growth rates. Simulations are performed in repeated batch conditions, where nutrients are replenished and

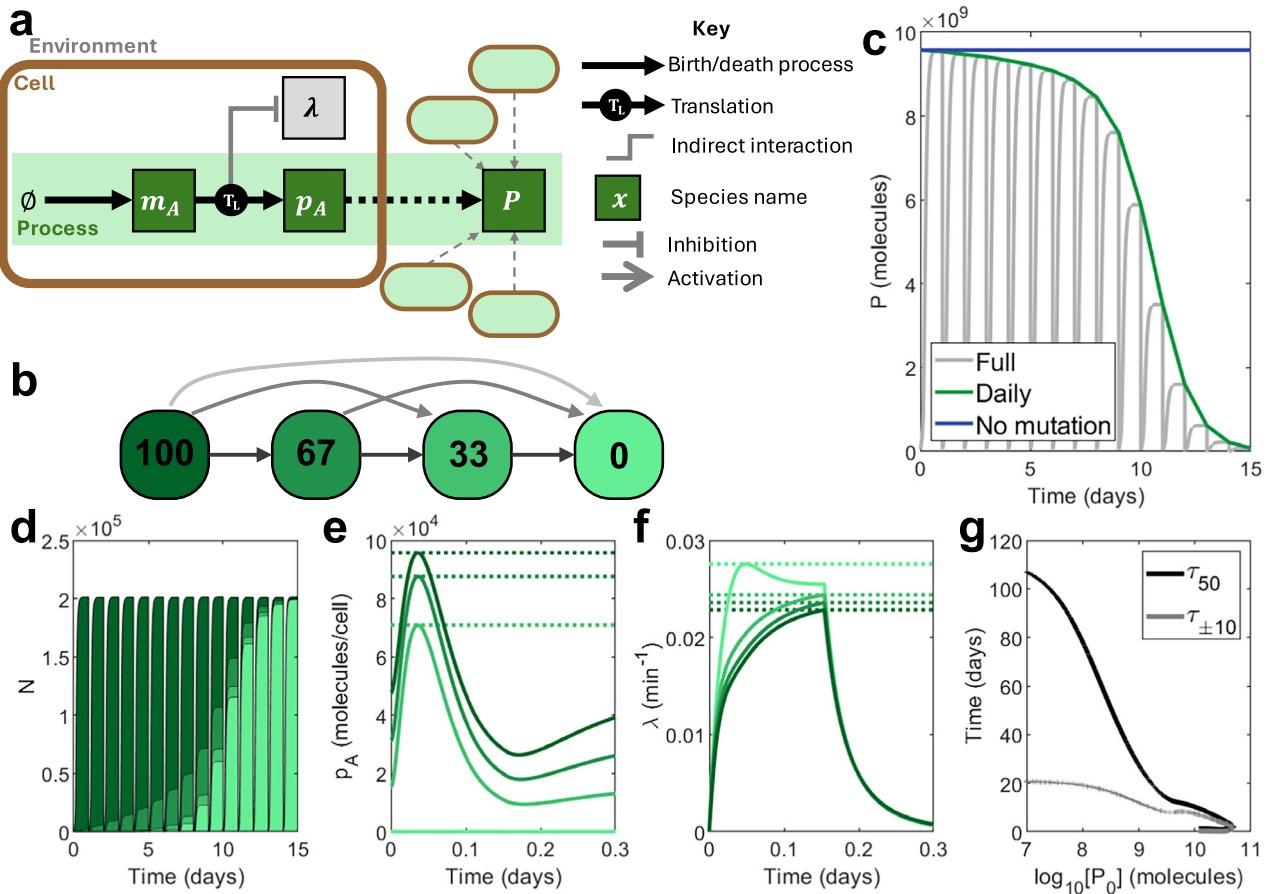

**Fig. 2 | Simulating an open-loop process in repeated batch conditions. a** A schematic showing the function of the process. $m_A$ is spawned and translated to create output $p_A$ using host ribosomes. This process impacts host growth. The total output $P$ comes from the output $p_A$ across an entire population of engineered cells. **b** A visual depiction of the mutation scheme for this process. Coloured squares represent distinct 'mutation states' with different levels of function. Numbers in the squares show the percentage function of a given state relative to the designed level, implemented through differences in the maximal transcription rate $\omega_A$: 100 represents a state functioning as designed, while 0 represents a completely non-functional state where no transcripts $m_A$ are produced. Arrows signify possible transitions between mutation states, with lighter arrows representing mutations which occur less often. **c–f** Time-series outputs using an open-loop process with maximal transcription rate $\omega_A = 5$ mc min$^{-1}$. **c** Total population-wide circuit output $P$, plotted both in full (grey) and at the end of each simulation day (green). An ideal system would match an open-loop system in the absence of mutation, maintaining function indefinitely (blue). **d** Population size $N$, distributed according to mutation state. Dark green represents a fully-functional (100%) strain. Light green represents a non-functional (0%) strain. **e** Output per cell $p_A$ according to mutation state over the first day. Dotted lines show maximum outputs. **f** Growth rate $\lambda$ according to mutation state over the first day. Dotted lines show maximum growth rates. **g** For a wide range of processes (varying $\omega_A$ between 0.1 and 1000 mc min$^{-1}$), $\tau_{\pm 10}$ (black) and $\tau_{50}$ (grey) against initial output $P_0$. Simulation results are provided as a Source Data file.

population size is reset every 24 hours, mirroring previous experiments[4,10]. Variables are given in molecules per cell (mc/cell). The complete model is defined in Supplementary Note 1. We perform simulations as defined in the Methods section entitled "Simulating an evolving population of engineered cells".

Here, we consider a simple synthetic process consisting of a single gene $A$, which could represent a fluorescent reporter protein such as GFP. Messenger RNA (mRNA) transcripts $m_A$ are generated proportional to the maximal transcription rate $\omega_A$. They combine with ribosomes $R$ supplied by the host model to form "translation complexes" $c_A$, which yield protein $p_A$ upon the completion of translation, having consumed cellular anabolites $e$. Through consumption of $e$ and utilisation of $R$ the host and circuit models become coupled and the phenomenon of burden is captured[32]. We define the total output of the system $P$ to be the total number of molecules of $p_A$ across the entire population composed of $i$ strains:

$$P = \sum_i (N_i p_{Ai}),\qquad(1)$$

where each strain $i$ represents a different mutant, and $N_i$ is the number of cells belonging to the $i$th strain (Fig. 2a). For this nominal open-loop system, we assume four distinct "mutation states" which differ in the value of the maximal transcription rate $\omega_A$, corresponding to 100%, 67%, 33% and 0% of the nominal or designed level. Mutation occurs via transition rates between these populations such that only function-reducing mutations may occur, and such that more extreme mutations are less likely (Fig. 2b). (See Supplementary Note 1 and Supplementary Fig. S1 for a full description of the mutation scheme.)

To measure the evolutionary longevity of this process, we define three metrics:

1. $P_0$, the initial output $P$ from the ancestral population prior to any mutation.
2. $\tau_{\pm 10}$, the time taken for the output $P$ to fall outside of the range $P_0 \pm 10\%$.
3. $\tau_{50}$, the time taken for the output $P$ to fall below $P_0/2$.

We define these metrics assuming that we wish to maximise production ($P_0$) and maintain performance near to the original state

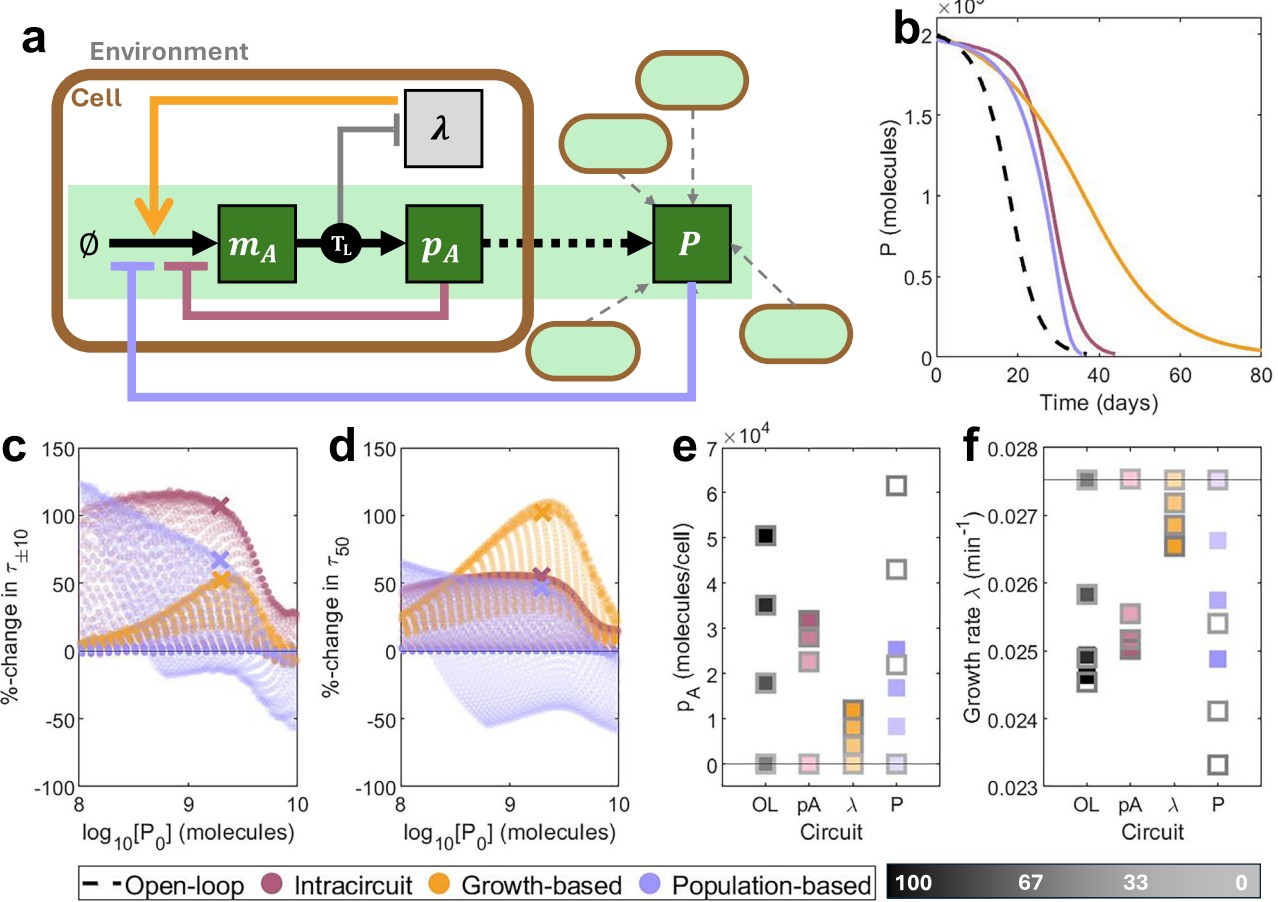

**Fig. 3 | Phenomenological modelling of different choices of control input. a** A simple schematic showing the three controller architectures: intra-circuit control (red), growth-based control (orange) and population-based control (lilac). Each output comes from a different layer of the combined multi-scale model (circuit model, host model and population model). See key in Fig. 2a for symbol meanings. **b** Time-series of population-wide output $P$ over time for an open-loop system (black, dashed, $w_A = 4.0$ mc min$^{-1}$) and representative control systems of equivalent initial output $P_0$. (Intra-circuit: red, $w_A = 10^3$ mc min$^{-1}$, $k_A = 4.5 \times 10^2$ mc. Growth-based: orange, $w_A = 87$ mc min$^{-1}$, $k_\lambda = 6.3 \times 10^{-2}$ min$^{-1}$. Population-based: lilac, $w_A = 16$ mc min$^{-1}$, $k_P = 5.2 \times 10^8$ mc.). **c**, **d** A large number of designs were generated by varying the maximal transcription rate $\omega_A$ and control parameters $k_u$. Against the initial output $P_0$, we plot the percentage change in (**c**) $\tau_{\pm 10}$ and (**d**) $\tau_{50}$ vs an open-loop system of equal initial output. Points marked with an X correspond to the time-series plots in (**b**). These were selected as points on the Pareto fronts simultaneously optimising $P_0$, $\tau_{50}$ and $\tau_{\pm 10}$, with initial output $P_0$ closest to $2 \times 10^9$ molecules. **e**, **f** For circuits corresponding to trajectories in (**b**), (**e**) maximum protein production per cell $p_A$ and (**f**) maximum growth rates $\lambda$ across the first day (solid) and the day where $\tau_{50}$ is reached (grey outline) for each mutation state. Lighter squares represent less functional strains. The horizontal line represents a non-functional strain. Simulation results are provided as a Source Data file.

($\tau_{\pm 10}$) for as long as possible. We propose $\tau_{50}$ as a measure of long-term performance or "persistence" because the maintenance of "some function" may be sufficient for many applications, such as biosensing.

For the proposed nominal open-loop system, the total output $P$ falls over the course of the simulation from its initial value $P_0$ to 0 molecules (Fig. 2c). This corresponds to a transition of the population makeup, which initially comprises entirely of fully functional cells but eventually is dominated by non-producing, faster growing mutants (Fig. 2d–f). For systems with increased process transcription (i.e., larger maximal transcription rates $\omega_A$), the initial output $P_0$ increases as more protein is produced per cell. However, this increases the burden caused by the process and therefore reduces both $\tau_{50}$ and $\tau_{\pm 10}$. Beyond a certain point, continuing to increase $\omega_A$ can overburden the cell, leading to a reduction in all metrics (Fig. 2g).

The objective of this paper is to evaluate the ability of different control strategies to improve the evolutionary longevity of this simple synthetic circuit. It is trivial to reduce the burden and improve both $\tau_{\pm 10}$ and $\tau_{50}$ by reducing the production of $p_A$ via the birth rate $\omega_A$. However, this impedes circuit function (Supplementary Fig. S2a). We therefore define the performance of controllers in comparison to an open-loop system of equal initial

output $P_0$. See Methods for Comparing controllers versus open-loop systems of equal output for full details as to how these metrics where determined. The ideal controller would extend $\tau_{\pm 10}$ and $\tau_{50}$ at no loss in $P_0$.

## Phenomenological models demonstrate the potential of negative feedback control to enhance evolutionary longevity

We consider three candidates for the choice of input for a control strategy designed to improve evolutionary longevity (Fig. 3a). We model all controllers phenomenologically by scaling the maximal transcription rate of the circuit gene $\omega_A$ by a regulatory function, so that protein production is inhibited when levels are already high:

$$w_A = \omega_A \cdot \Theta(u). \tag{2}$$

Here, $u$ is the control input of choice.

Firstly, we consider product-based negative feedback where the protein per cell $p_A$ is the control input (Fig. 3a, red). To maintain function close to a desired level, production is inhibited if there is an abundance of output and control is alleviated if production is low. We

call this intra-circuit feedback. This is achieved via the regulatory function:

$$\Theta(p_A) = \frac{k_A{}^2}{k_A{}^2 + p_A{}^2}. \tag{3}$$

Should a mutation occur which reduces the production of $p_A$, the strength of control will be eased, allowing the actual output levels to rise again. This means that mutations which reduce circuit output will have a lesser impact on burden, minimising the selective advantage of such mutations.

Secondly, we consider growth rate $\lambda$ as the control input (Fig. 3a, orange). Since increased production of $p_A$ corresponds to reduced growth, we employ feedback so that production is inhibited at low growth. We call this growth-based feedback, and use the regulatory function:

$$\Phi(\lambda) = \frac{\lambda^2}{k_\lambda{}^2 + \lambda^2}. \tag{4}$$

If a burden-relieving mutation arises with reduced production of output $p_A$, control will be alleviated, enabling a resultant increase in output and a smaller selective advantage over the original designed strain.

Finally, since our objective is to maintain the performance of a synthetic circuit across an entire population, we consider using the total population-wide output $P$ as an input for control (Fig. 3a, lilac). Such a system could be implemented physically using cell-to-cell communication systems such as quorum sensing. We employ feedback so that further production is inhibited when the population-wide output $P$ is high. We call this population-based feedback, and use the regulatory function:

$$\Theta(P) = \frac{k_P{}^2}{k_P{}^2 + P^2}. \tag{5}$$

Unlike with the other systems, individual cells do not directly respond to mutations in the circuits they carry by alleviating control, because such mutations do not significantly alter the population-level control input $P$. Instead, this system aims to maintain population-level output by steadily increasing the per-cell production of non-mutant cells to compensate for the increasing number of non-functional mutants.

For all three systems, applying control to a given nominal process (here, $\omega_A = 10$ mc min$^{-1}$) boosts longevity at the expense of initial output $P_0$, with stronger controllers (low $k_A$, high $k_\lambda$, low $k_P$) causing a more significant change (Supplementary Fig. S2b–d). Representative dynamics from systems which initially produce the same population-wide output $P$ are shown in Fig. 3b. This initial comparison suggests that intra-circuit control is capable of maintaining function at close to the designed level for longer (greater $\tau_{\pm10}$), but that growth-based control is more effective at improving long-term persistence (greater $\tau_{50}$). To assess the different strategies at the topological level, we generated many designs by varying both the process input $\omega_A$ and the controller strength ($k_A$, $k_\lambda$ and $k_P$) while all other system parameters were fixed as defined in Supplementary Table 4. We observe that intra-circuit control excels in the short term while growth-based control excels in the long term across most parameter combinations, demonstrating that the result holds at the topological level (Fig. 3c, d). Except in cases of minimal output, population-based control is less effective and is the only strategy where some designs perform worse than open-loop. These differences in performance are a result of differences in the per-cell protein production and growth rates between mutant strains (Fig. 3e, f). For intra-circuit control, mutating into intermediate mutation states (where function is non-zero but less than

the designed level) alleviates some of the control exerted by the controller, pushing production back up and providing a smaller growth advantage. However, this means that mutations which completely abolish function provide a greater growth advantage, allowing non-functional mutants to dominate quickly, accelerating loss-of-function. Growth-based control is capable of boosting the growth rate of the fully-functional state, thereby reducing the selective advantage of mutation into any other state. For population-based control, non-mutant cells become affected by a decline in population-wide output, so that control strength is eased over time. This allows them to produce more output, but leads to an even greater selective disadvantage, thereby accelerating loss-of-function.

Having established the theoretical differences in performance for different controller input strategies, we next consider how such systems could be implemented in living cells by deriving mechanistic models of biologically realisable topologies. We consider both transcription factor-based and RNA-based methods of control information processing and actuation. In the main text, we report the results for the intra-circuit and growth-based systems. Mechanistic modelling of a population-based controller which utilises quorum sensing confirms that such systems have poor performance (Supplementary Figs. S3–S6). These results are described in Supplementary Note 2.

### Intra-circuit feedback control improves short-term performance at the expense of longevity

The intra-circuit control strategy described in the above section and demonstrates the potential of negative feedback control to enhance evolutionary longevity (Fig. 3) is equivalent to an auto-regulation motif where an inhibitory transcription factor inhibits itself. This limits its utility in practice, where one may want to stabilise the expression of an enzyme or signalling protein that performs a useful circuit-specific function. Here, we consider two biologically implementable control systems where feedback is enacted in proportion to the per-cell protein output $p_A$. The first is composed of an inhibitory transcription factor $p_B$, which is produced from the same promoter as circuit protein $p_A$, so that the functional identity of $p_A$ is not constrained. $p_B$ inhibits the production of both $p_A$ and $p_B$ (Fig. 4a). In this way, when production is high, increased inhibition reduces the production of new synthetic proteins, enabling the reduction of burden. This approach imitates previously implemented controllers (such as ref. 26). We call this controller CLpATX ("Closed-loop, senses $p_A$, prevents transcription"). To implement this, we scale the birth rate of mRNAs by the regulatory function:

$$\Theta_A(p_B) = \frac{k_B{}^2}{k_B{}^2 + p_B{}^2}.$$

Note that the controller protein here will exert an additional burden on top of the process gene.

The second negative feedback mechanism similarly makes use of a co-produced transcription factor $p_B$, but in this case it activates an sRNA $r_C$ which then eliminates the mRNA $m_A$ by sequestration (as in e.g.[35]) (Fig. 4b). We call this controller CLpATL ("Closed-loop, senses $p_A$, prevents translation"). (For ease of reference, the names of the controllers, their inputs and mode of action are available in Supplementary Table S1). Note that there are now two genes which may be subject to mutation (the process and the sRNA) and so we update our mutation model to include $4^2 = 16$ distinct mutation states, where each represents a unique pair [$\%_A$, $\%_C$], with $\%_A$, $\%_C \in \{100\%, 67\%, 33\%, 0\%\}$. We assume that mutations only affect a single promoter at a time (Supplementary Fig. S1). (See Supplementary Note 1 for a full description of both models for a full description of the mutation scheme.)

We first consider the transcriptional controller architecture: CL$p_A$TX. To determine its performance across a range of processes, we

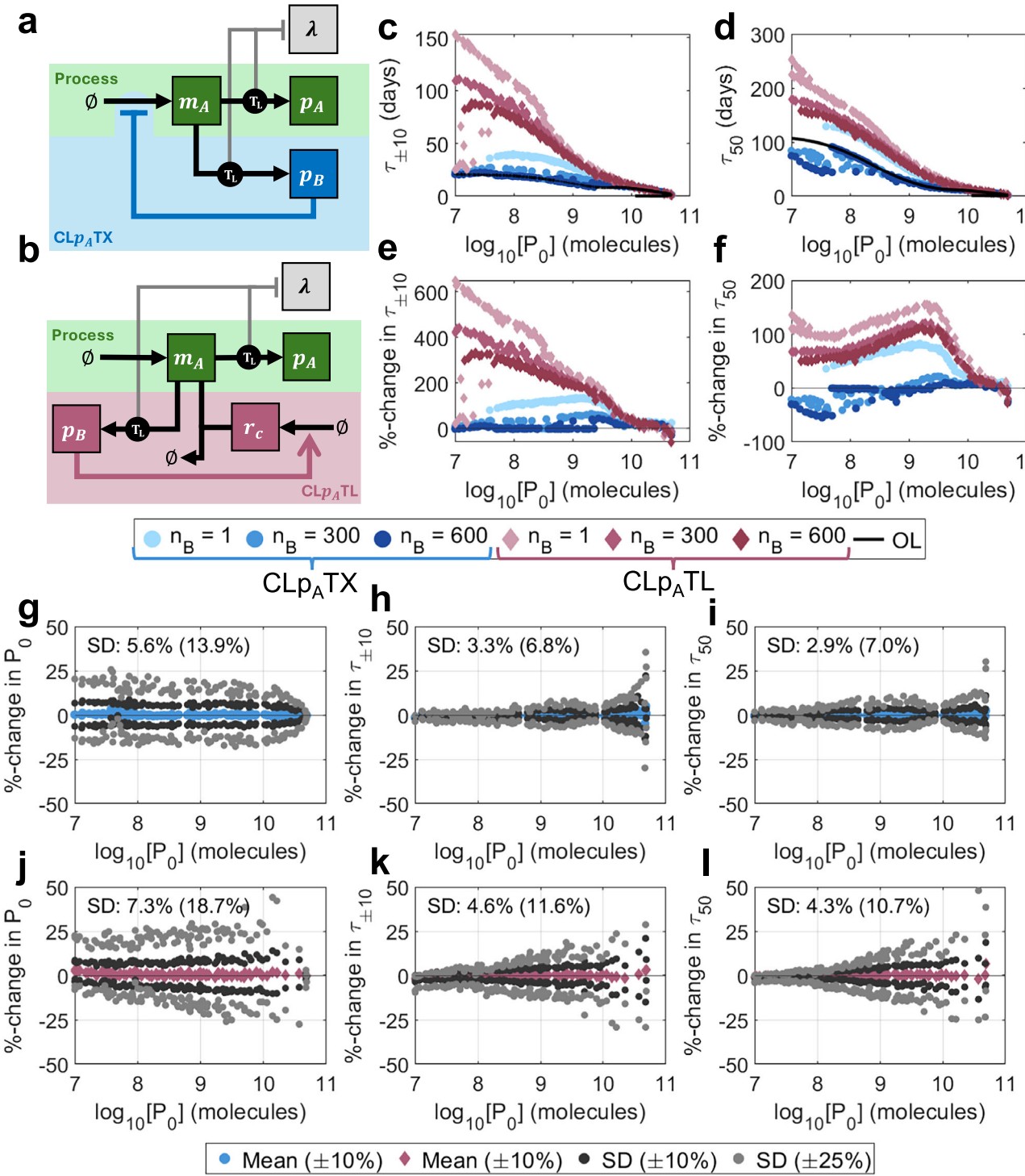

**Fig. 4 | Protein-based control design and performance. a** A schematic describing the controller CL$p_A$TX. Controller protein $p_B$ is produced from the same gene as output protein $p_A$ and acts as an inhibitory transcription factor on the shared gene. **b** A schematic describing the controller CL$p_A$TL. See key in Fig. 2a for symbol meanings. A controller protein, $p_B$, is produced from the same gene as the output protein $p_A$, and acts as a transcription factor which activates the production of sRNA $r_C$. The feedback loop is completed through the combining of $r_C$ with mRNA that codes for $p_A$ and $p_B$, preventing it from being translated. **c**–**f** Optimal performance for both CL$p_A$TX (blue) and CL$p_A$TL (red) for controller protein length $n_B = 1, 300, 600$ aa. **c** $\tau_{\pm 10}$ vs initial output $P_0$, (**d**) $\tau_{50}$ vs initial output $P_0$, (**e**) %-change in $\tau_{\pm 10}$ over open-loop vs initial output $P_0$, (**f**) %-change in $\tau_{50}$ over open-loop vs initial output $P_0$. **g**–**l** Robustness analyses for (**g**–**i**) CL$p_A$TX and (**j**–**l**) CL$p_A$TL. For each of the 100 optimal controllers with $n_B = 300$ aa, 100 further controllers were generated by varying parameters by up to $\pm 10\%$ (dark grey) and $\pm 25\%$ (light grey). The percentage changes in three output metrics were calculated versus the original optimal systems: (**g**, **j**) $P_0$, (**h**, **k**) $\tau_{\pm 10}$ and (**i**, **l**) $\tau_{50}$. Plots show the means (for $\pm 10\%$) and standard deviations (for both $\pm 10\%$ and $\pm 25\%$) of the percentage changes for each optimal controller. Percentages marked on the plots indicate the standard deviations across the entire Pareto front when parameters were varied by $\pm 10\%$ ($\pm 25\%$). Simulation results are provided as a Source Data file.

performed multi-objective optimisations to simultaneously maximise $P_0$, $\tau_{\pm 10}$ and $\tau_{50}$, varying the control strength $k_B$, the control protein ribosome binding rate $b_B$ and the circuit maximal transcription rate $\omega_A$ (see Methods section "Multi-objective optimisation"). To evaluate performance, we then compared these outputs with corresponding open-loop systems of equal initial output $P_0$. (See Methods section "Comparing controllers versus open-loop systems of equal output").

For controller protein lengths typical of bacterial regulators (i.e., $n_B = 300$ amino acids (aa)), improvements in $\tau_{50}$ are minimal, but $\tau_{\pm 10}$ can be improved by up to 60% versus open-loop (Fig. 4c–f, blue). Our optimisations show that, where improvements are possible, control strength should be maximised (i.e., low $k_B$ across the Pareto front (Supplementary Fig. S7, blue). This is because stronger inhibitors do not need to be as abundant to achieve the same strength of feedback, and so controller burden can be minimised. At low outputs $P_0$, where performance increases are minimal or non-existent, controller binding strength $b_B$ is very small, so there is less pressure to minimise $k_B$ as controller proteins are less abundant. Systems which produce less output (and have increased longevity) tend to have less synthetic transcription (low $\omega_A$) and less controller translation (low $b_B$) (Supplementary Fig. S7, blue).

We tested the robustness of the optimal designs to parametric uncertainty by varying the optimal parameters randomly by up to $\pm 10\%$ to generate 10,000 alternate designs. We repeated the analysis for $\pm 25\%$. To quantify robustness, we tracked the percentage change in each metric ($P_0$, $\tau_{\pm 10}$ and $\tau_{50}$) versus its corresponding optimal system. A robust architecture would have small standard deviations in these values when the designs across the Pareto front are subjected to variation (See Methods section "Robustness analysis"). Here, optimal controllers are very robust to parametric variation, with changes in $P_0$, $\tau_{\pm 10}$ and $\tau_{50}$ having standard deviations of 5.6% (13.9%), 3.3% (6.8%), and 2.9% (7.0%) respectively, compared with the original optimal systems when parameters were varied by $\pm 10\%$ ($\pm 25\%$) (Fig. 4g–i). (See Supplementary Note 3 for our comprehensive robustness analysis (Supplementary Fig. S8a–d and Supplementary Table S2).)

In an ideal case where the repressor causes minimal burden (i.e., $n_B \rightarrow 0$ aa), performance can be significantly boosted to the levels suggested by the phenomenological models. The binding rate $b_B$ can be increased without significantly increasing burden (Supplementary Fig. S7, blue). However, increasing the repressor size $n_B$ to 600 aa, and therefore increasing controller burden, performance deteriorates to worse than open-loop and any benefits of control are lost entirely. As repressor size increases from 300 aa to 600 aa, the parametric design rules do not qualitatively change (Supplementary Fig. S7, blue).

To better understand how design choices impact this controller architecture, we tested the performance of this controller for a nominal process of maximal transcription rate $\omega_A = 50$ mc min$^{-1}$ and controller protein length $n_B = 300$ aa by varying the controller ribosome binding rate $b_B$ and the controller strength $k_B$. We show that $\tau_{\pm 10}$ can be improved by a more significant margin than $\tau_{50}$, and across a wider portion of the design space, suggesting that it is easier to boost short-term performance than long-term performance (Supplementary Fig. S9). (See Supplementary Note 4.)

## Post-transcriptional control outperforms transcriptional control for intra-circuit feedback

Unlike CL$p_A$TX, which only enhances performance when controller size (and therefore burden) is minimal, CL$p_A$TL is capable of significantly improving both $\tau_{\pm 10}$ and $\tau_{50}$, even in the presence of large controller sizes ($n_B = 600$ aa) (Figs. 4c–h, 5a, red). For typical regulator lengths ($n_B = 300$ aa), $\tau_{50}$ can be more than doubled, with $\tau_{\pm 10}$ showing an even more extreme improvement at low initial outputs of up to +400% versus open-loop. Although this system significantly

outperforms CL$p_A$TX, the optimal control designs are less robust, with random parameter variation of up to $\pm 10\%$ ($\pm 25\%$) causing larger variations in output metrics: changes in $P_0$, $\tau_{\pm 10}$ and $\tau_{50}$ have standard deviations of 7.3% (18.7%), 4.5% (11.6%) and 4.3% (10.7%) respectively (Fig. 4j–l). (See Supplementary Note 3 for a comprehensive robustness analysis (Supplementary Fig. S8e–h and Supplementary Table S2).) As with CL$p_A$TX, maximising controller strength (i.e., minimising $k_B$) is crucial to achieve the most control for the least burden for $n_B = 300$ aa and $n_B = 600$ aa. For $n_B = 1$ aa, controller burden is much less significant, so pushing $k_B$ to its absolute minimum is less important. Producing sRNA $r_C$ does not result in an additional burden, and so maximising its production allows for the strongest control (Supplementary Fig. S7, red). The demonstrated improvements in longevity of CL$p_A$TL over CL$p_A$TX are a result of two key mechanisms: (i) reduced controller burden (at equivalent control strength) and (ii) the emergence of mutations to the controller itself (enabling the short-term growth of high-producing strains).

Firstly, we see that optimal controllers utilise larger maximal transcription rates $\omega_A$ (Fig. 5b) and require much less controller protein $p_B$ than CL$p_A$TX (Fig. 5c). This is a result of the $p_B$-mediated activation of $r_C$, which amplifies the "impact" that each burdensome controller protein can achieve. This enables CL$p_A$TL to provide "more control for less burden" and significantly enhance performance in the long term.

Secondly, this process-controller system has two promoters ($A$ driving the gene expression and $C$ driving the controller). When the process promoter mutates (affecting the process transcription rate $\omega_A$), synthetic protein production falls, causing an increase in growth rate. However, when the sRNA promoter mutates (affecting the sRNA transcription rate $\omega_C$), the strength of control falls, so synthetic protein production rises, leading to a decrease in growth rate. Over time, the emergence of both types of mutation leads to a heterogeneous population of mutant strains, some of which produce more protein $p_A$ than the ancestral strain (Fig. 5d–f). It is important to note that there are two sources of mutant spread: (1) the emergence of new mutants via random mutation and (2) the competition between mutant strains as a result of their relative growth rates. In the long term, growth rate differences are the predominant cause of mutant domination; the higher-producing strains are outcompeted by faster-growing strains. However, early in the time course, when the majority of the population is still fully functional, the dynamics is primarily affected by the emergence of new mutations. Mutations to the controller can therefore initially balance out with mutations to the process, enabling function to be maintained close to the designed level for much longer, until out-competition becomes the dominant source of mutant spread. This effect also explains why the improvement in $\tau_{\pm 10}$ is so high for smaller initial outputs $P_0$: the differences in growth rate between mutants become much smaller, so the emergence of new mutants makes up a bigger proportion of the mutant spread, so cells with mutated controllers (and increased output) make up a larger fraction of the combined population, balancing out the cells with reduced process output. This results in a linear relationship between $\tau_{\pm 10}$ and $\tau_{50}$ for optimal systems as the initial output $P_0$ varies. This contrasts with CL$p_A$TX, where increases in $\tau_{\pm 10}$ are more restricted than increases in $\tau_{50}$ at low outputs $P_0$ (Supplementary Fig. S10a, e).

These higher-producing mutants are only prevalent in the short term but contribute significantly to the excellent short-term performance ($\tau_{\pm 10}$) of CL$p_A$TL. The improvement in long-term performance over CL$p_A$TX is primarily a result of the reduced burden. To verify this claim, we separately considered a system with two dimensions of mutation but no additional burden. We observe improvements in short-term performance when we include a second controller-specific mutation site whose mutation leads to an increase in process output. However, long-term performance, optimal parameter choices and robustness are not significantly impacted (Supplementary Figs. S11 and S12). (See full details in Supplementary Note 5.)

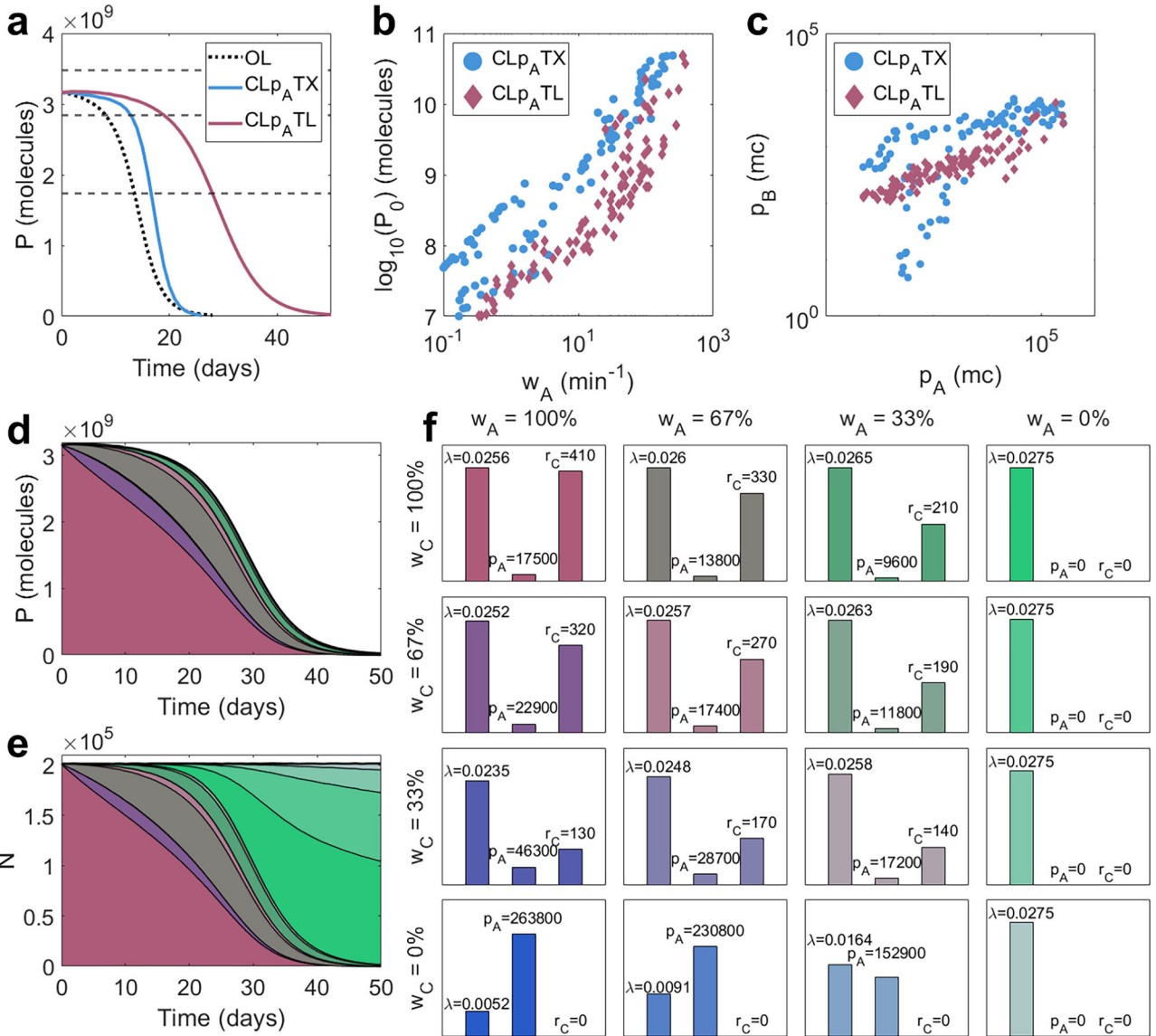

**Fig. 5 | Understanding the improved performance of CL$p_A$TL. a–c** Comparing CL$p_A$TX (blue) and CL$p_A$TL (red). **a** Population-wide process outputs $P$ over time for two representative controllers. **b** Initial output $P_0$ against the maximal transcription rate of the process gene $\omega_A$ across all optimal controllers. **c** Controller protein per cell $p_B$ vs process output protein per cell $p_A$ across all optimal controllers. **d–f** The effect of controller mutation on CL$p_A$TL with $n_B = 300$ aa. **d** Population-wide output

$P$ over time according to mutation state. **e** Population size according to mutation state. **f** Bar charts for CL$p_A$TL with $n_B = 300$ aa showing (i) growth rate $\lambda$ (min$^{-1}$), (ii) process protein output per cell $p_A$ (molecules/cell) and (iii) sRNA per cell $r_C$ (molecules/cell). Each subplot represents a unique mutation state. Moving rightwards signifies mutation of the process, and moving downwards signifies mutation of the controller. Simulation results are provided as a Source Data file.

As with CL$p_A$TX, we explored how design choices influence performance for a nominal process ($\omega_A = 50$ mc min$^{-1}$). As before, we see that $\tau_{\pm10}$ improves over a larger portion of the design space than $\tau_{50}$, demonstrating that short-term maintenance of function is easier to improve than long-term persistence (Supplementary Figs. S13 and S14). See Supplementary Note 4 for a full description.

We have shown that CL$p_A$TL outperforms CL$p_A$TX partly as a result of the low burden of transcription compared with translation. We extended our cell model to capture energy consumption, and therefore burden, by transcription. We found that $CLp_ATX$ is insensitive to this additional burden and that $CLp_ATL$ outperforms it except at the highest transcriptional burden (Supplementary Fig. S15). See Supplementary Note 6 for a comprehensive discussion. Given this observation, we proceed with the assumption that transcriptional burden remains negligible and that translational demands (in terms of both energy consumption and cellular resource competition) dominate

bacterial growth and host-circuit interactions, in accordance with established models and experimental data[19,31,36–38].

## Growth-based feedback enhances evolutionary longevity

Synthetic circuit burden elicits the upregulation of specific natural promoters, which can be considered to be burden sensors. These promoters can be exploited to design control systems that are sensitive to burden[39]. When burden is high and growth is low, greater restriction on the process can alleviate burden and reduce the selective benefit of mutations to the process gene. As with intra-circuit control, we considered two growth-based controller designs: CL$\lambda$TX ("Closed-loop, senses $\lambda$, prevents transcription") (where control is exerted by an inhibitory transcription factor) and CL$\lambda$TL ("Closed-loop, senses $\lambda$, prevents translation") (which exploits sRNA-mediated mRNA sequestration). Both CL$\lambda$TX and CL$\lambda$TL improve evolutionary performance, with CL$\lambda$TL outperforming CL$\lambda$TX in both the short term (greater $\tau_{\pm10}$) and long term (greater $\tau_{50}$ except at very high initial

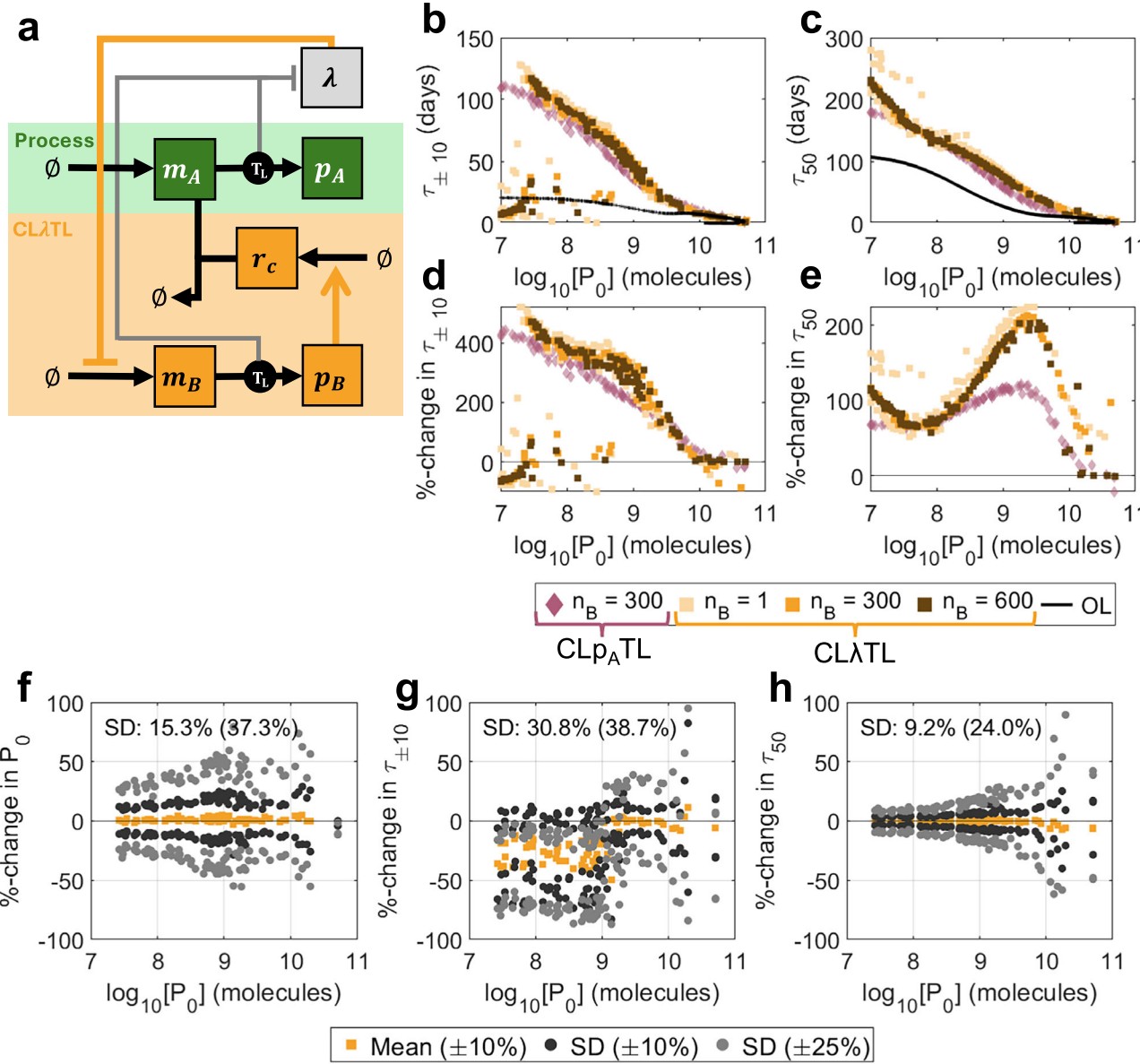

**Fig. 6 | Optimal outputs for CLλTL. a** A schematic describing the controller. Controller protein $p_B$ is produced from a growth-sensitive promoter and activates the production of sRNA $r_C$. $r_C$ combines with and deactivates process mRNA $m_A$. See key in Fig. 2a for symbol meanings. **b–e** Optimal performance for both CLλTL (orange) and CL$p_A$TL (red) for controller protein length $n_B = 1, 300, 600$ aa. **b** $\tau_{\pm10}$ vs initial output $P_0$, (**c**) $\tau_{50}$ vs initial output $P_0$, (**d**) %-change in $\tau_{\pm10}$ over open-loop vs initial output $P_0$, (**e**) %-change in $\tau_{50}$ over open-loop vs initial output $P_0$. **f–h** Robustness analysis. For each of the 100 optimal controllers with $n_B = 300$ aa,

100 further controllers were generated by varying parameters by up to ±10% (dark grey) and ±25% (light grey). The percentage changes in three output metrics were calculated versus the original optimal systems: (**f**) $P_0$, (**g**) $\tau_{\pm10}$ and (**h**) $\tau_{50}$. Plots show the means (for ±10%) and standard deviations (for both ±10% and ±25%) of the percentage changes for each optimal controller. Percentages marked on the plots indicate the standard deviations across the entire Pareto front when parameters were varied by ±10%(±25%). Only original optimal controllers where $\tau_{\pm10} = \tau_{90}$ are considered. Simulation results are provided as a Source Data file.

outputs $P_0$) (Supplementary Fig. S16). Here, we focus our analysis on CLλTL, with the discussion of CLλTX presented in Supplementary Note 7 (Supplementary Figs. S16 and S17). This controller exploits growth-sensitive promoters to drive transcription of an sRNA $r_C$ (Fig. 6a). When the cell undergoes stress (i.e., low growth rate), a controller protein $p_B$ is produced, which activates sRNA expression ($r_C$). This sRNA combines with and eliminates the mRNA for the process gene $m_A$, preventing it from being translated. There are three promoters in this system, so we employ a mutation scheme consisting of $4^3 = 64$ distinct strains corresponding to unique triplets {%$_A$, %$_B$, %$_C$} with {%$_A$, %$_B$,%$_C$} ∈ {100, 67, 33, 0} (Supplementary Fig. S1). (See Supplementary Note 1 and Supplementary Fig. S1 for a full description of the mutation scheme.)

For typical regulator protein sizes ($n_B = 300$ aa), CLλTL slightly outperforms CL$p_A$TL in the short term, but significantly outperforms it in the long term, with possible improvements of more than 200% versus open-loop (Figs. 6b–e, 7a). The difference in performance in systems with minimal burden ($n_B = 1$ aa) and high burden ($n_B = 600$ aa) is very small, suggesting that this controller topology is less sensitive to controller burden than intra-circuit controllers. Observing the population dynamics, we see that when mutations affect the process gene, there is less variation in protein production and a minimal loss in growth rate, so mutant strains with reduced function have less of a selective advantage (Fig. 7b, d). Similarly, when mutations affect the controller, the new mutant strains with increased function have less of a selective disadvantage (Fig. 7c, e).

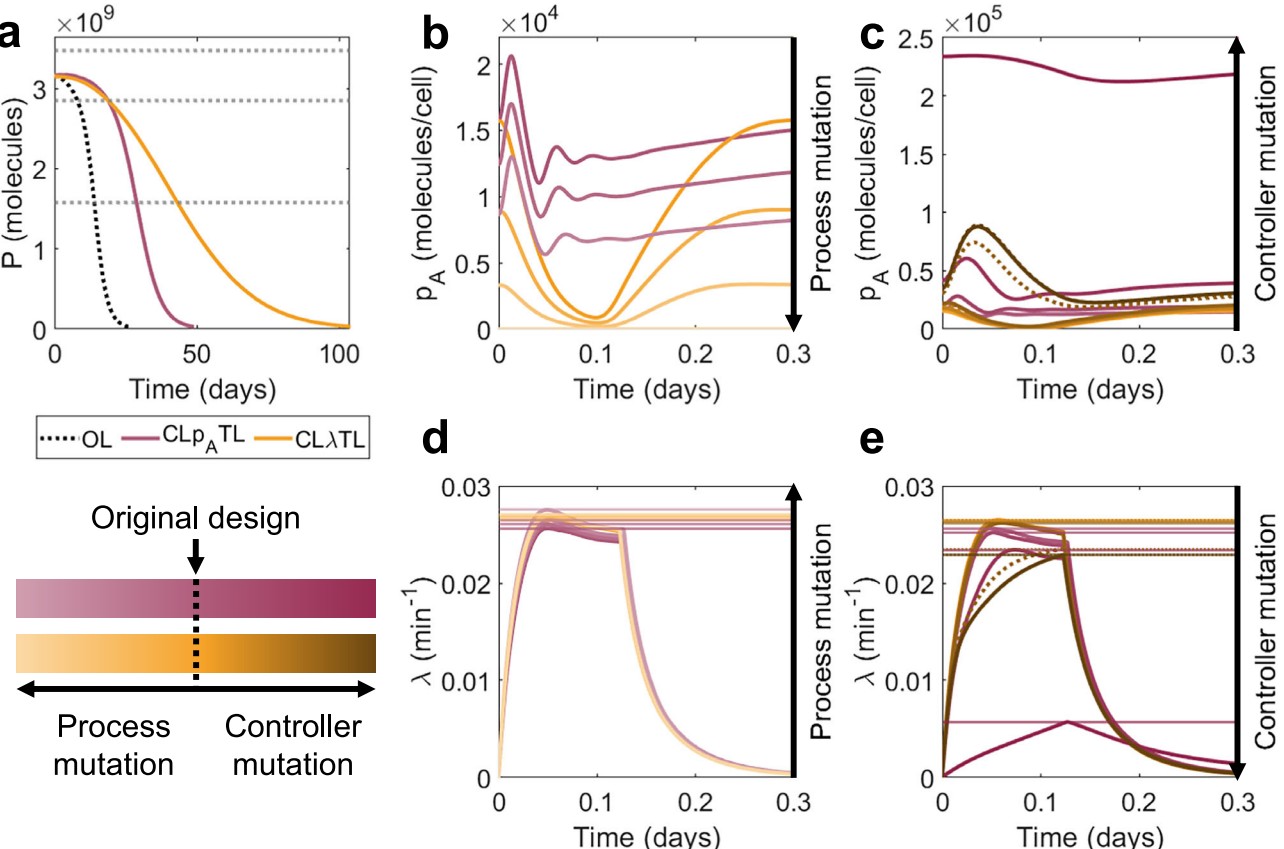

**Fig. 7 | Why does CL$\lambda$TL outperform CL$p_A$TL? a** Time-series output for representative optimal controllers with $n_B = 300$ aa. (Open-loop black dotted line). **b** Protein output per cell $p_A$ over time for the first day of simulation according to mutation state, considering only states with a mutated process and a fully functional controller. **c** Protein output per cell $p_A$ over time for the first day of simulation, considering only states with a mutated controller and fully functional process (solid line mutation in $\omega_B$, dotted line mutation in $\omega_C$). **d, e** Growth rates $\lambda$ according to mutation state. Main plots show time series over the first day. Horizontal lines indicate differences in the maximum growth rates of each state over the first day. **d** Only states with mutated process. **e** Only states with mutated controllers (solid line/square is mutation in $\omega_B$, dotted line/triangle is mutation in $\omega_C$). Simulation results are provided as a Source Data file.

Investigating the optimal controller designs shows only $\omega_A$ has significant variation across the Pareto front. All other parameters demonstrate clear optimal choices across the performance space (Supplementary Fig. S17, orange). The strength of control from the transcription factor should be maximised ($k_B$ minimised). This becomes more crucial as protein length increases, corresponding also to a reduction in ribosome utilisation (reduced $b_B$) but increased transcription of controller genes (increased $\omega_B$ and $\omega_C$). $k_A$ controls the sensitivity of the controller to changes in growth rate. Our optimisations show that controllers should have $k_\lambda$ in the region of 0.002 to 0.005 min$^{-1}$, suggesting a high sensitivity to growth rate.

We note that some optimal systems with low initial outputs have $\tau_{\pm10}$ worse than open-loop. Investigating the system dynamics shows that these "fail" because their output increases significantly in the early stages of the simulation as a result of controller mutation (Supplementary Fig. S18a–c). These are systems for which the 90%-life $\tau_{90}$ (defined as the time taken for output to fall below 90% of its original value) is not equal to $\tau_{\pm10}$. These systems have low output and very strong controllers, so when the controller mutates, they produce a comparatively very large amount of output protein. Despite these strains with mutant controllers having very low growth rates and only arising in small numbers, their contribution to the population-wide output is significant (Supplementary Fig. S18e). The effect is enhanced by the existence of two separate controller genes, both of which can be mutated to yield this increase in output. This effect can be exploited to generate systems with excellent $\tau_{50}$ but poor $\tau_{\pm10}$ because of an initial

increase in output before an eventual fall. This effect is more pronounced in systems with very low initial outputs $P_0$, because the functional systems are less burdensome and therefore have less of a growth disadvantage (Supplementary Fig. S18f, g). This means that a greater proportion of mutant spread initially comes from the emergence of new mutants rather than competition between strains.

This phenomenon also makes this topology much less robust to parametric variation. In some cases, small variations in parameters can significantly and discretely alter the value of $\tau_{\pm10}$ as systems which previously remained below +10% of the original value may now exceed it. Considering only the original designs where $\tau_{\pm10} = \tau_{90}$, 21.6% (28.2%) of designs fail to maintain $\tau_{\pm10} = \tau_{90}$ after parameters are randomly varied by up to ±10% (±25%). Standard deviations in the percentage change of $P_0$, $\tau_{\pm10}$ and $\tau_{50}$ are 15.3% (37.3%), 30.8% (38.7%) and 9.2% (24.0%), demonstrating that, despite the improvements in long-term performance, this controller topology is much less robust than intra-circuit topologies (Fig. 6f–h). As with the intra-circuit systems, the post-transcriptional feedback mechanism (CL$\lambda$TL) enables much greater optimal performance than the transcriptional feedback mechanism (CL$\lambda$TX), but this comes at the expense of a reduction in robustness: for CL$\lambda$TX, standard deviations in the percentage change of $P_0$, $\tau_{\pm10}$ and $\tau_{50}$ are 9.1% (22.6%), 22.7% (35.9%) and 9.2% (17.4%) (Supplementary Fig. S16h–j). (See Supplementary Note 3 for a detailed robustness analysis of both controllers (Supplementary Fig. S19).) For systems where $\tau_{\pm10} = \tau_{90}$, $\tau_{\pm10}$ and $\tau_{50}$ maintain a linear relationship (Fig. S10b, e).

To understand how design choices influence the performance of growth-based feedback systems, we considered how varying the controller parameters would influence the performance of CL$\lambda$TX for a nominal process of $\omega_A = 50$ mc min$^{-1}$. This is described in Supplementary Note 4. Unlike intra-circuit feedback, where it is easier to improve $\tau_{\pm10}$, growth-based feedback improves $\tau_{50}$ across a wider portion of the design space, suggesting that growth-based controllers could have greater potential for in vivo implementation in applications where circuit persistence is more important for maintenance of function close to the designed level (Supplementary Fig. S20).

We extended our analysis of growth-based control by considering a protein-free implementation where the growth-sensitive promoters drive the sRNA expression directly (rather than via a regulator protein). This system can improve performance versus open-loop but does not outperform CL$\lambda$TL (Supplementary Figs. S21 and S22). See Supplementary Note 8.

## Combining control schemes enhances robustness of growth-based feedback

We have shown intra-circuit control excels at maintaining short-term performance because it reduces the selective advantage of intermediate mutation states, whereas growth-based control excels at enhancing long-term performance because it boosts the growth rate of the fully functional state. Further, we have shown that sRNA-mediated controllers have enhanced optimal performance but poorer robustness than controllers which actuate directly via transcription factors. We therefore considered whether combining multiple inputs and mechanisms of feedback could combine the benefits of the individual systems and enable greater improvements in both longevity and robustness. We first considered a phenomenological model of a control strategy which utilises both product-based and growth-based control in conjunction. This system is described in Supplementary Note 9 and is capable of combining the benefits of both intra-circuit feedback and growth-based feedback to significantly enhance both short-term and long-term performance by up to 50% compared with the best performing single-input systems (Supplementary Fig. S23).

To assess the impact of biochemical constraints and controller burden on this strategy, we developed five multi-input mechanistic models which exploit both transcriptional and translational mechanisms of introducing feedback control: (i) CL$p_A$TL$\lambda$TL uses protein-based feedback to inhibit transcription of the process gene and growth-based feedback to prevent translation of the process gene by activating sRNA (Fig. 8a), (ii) CL$p_A$TL$\lambda$TX uses protein-based feedback to prevent translation of the process gene by activating sRNA and growth-based feedback to inhibit transcription of the process gene (Fig. 8b), (iii) CL$p_A$TL$\lambda$TL uses both protein-based feedback and growth-based feedback to co-operatively activate sRNA production and prevent translation of the process gene (Fig. 8c), (iv) CL$p_A$TX$\lambda$TX uses both protein-based feedback and growth-based feedback to co-operatively inhibit transcription of the process gene (Supplementary Fig. S24a) and (v) CL$p_A$TL$\lambda$PF (PF := protein-free) uses both protein-based feedback and growth-based feedback to co-operatively prevent translation of the process gene by putting the sRNA gene directly on a growth-sensitive promoter (Supplementary Fig. S24b). Models for all five multi-input controllers are detailed in Supplementary Note 1.

Here, we present a discussion of systems (i–iii). We solved a multi-objective optimisation problem to identify Pareto optimal designs and found that the potential improvements gained from combining the control inputs are counteracted by additional controller burden, leading to only marginal improvements in performance versus CL$\lambda$TL (Fig. 8d–g). The three multi-input controllers perform very similarly and obey similar primary qualitative design principles: maximise the production of sRNA (large $w_C$), minimise the ribosome binding rate of the growth-sensitive protein $p_{B_2}$ (small $b_{B_2}$), and maximise the

strength of transcriptional control (small $k_{B_1}$ and $k_{B_2}$) (Supplementary Fig. S25). The relationship between $\tau_{\pm10}$ and $\tau_{50}$ is the same as that for CL$\lambda$TL (Supplementary Fig. S10c, f). CL$p_A$TL$\lambda$TX showed the best robustness overall, with changes in $P_0$, $\tau_{\pm10}$ and $\tau_{50}$ having the standard deviations: 8.1% (20.0%), 16.9% (24.6%) and 6.8% (14.3%) (compared with 15.3% (37.3%), 30.8% (38.7%) and 9.2% (24.0%) for CL$\lambda$TL) when parameters were randomly varied by up to $\pm10\%$ ($\pm25\%$) (Fig. 8h–j). Further, only 8.8% (13.4%) of controllers failed to maintain $\tau_{\pm10} = \tau_{90}$ (compared with 21.5% (28.2%) for CL$\lambda$TL). Both CL$p_A$TX$\lambda$TL and CL$p_A$TL$\lambda$TL also show significantly improved robustness compared with CL$\lambda$TL (Fig. S26).

Although CL$p_A$TX$\lambda$TX enhances performance versus open-loop, it performs worse than systems (i–iii) without providing a notable improvement in robustness: changes in $P_0$, $\tau_{\pm10}$ and $\tau_{50}$ have the standard deviations 11.1% (28.1%), 19.1% (28.6%) and 4.2% (10.1%) (Supplementary Fig. S24, navy). 80.6% of controllers retain $\tau_{\pm10} = \tau_{90}$. At the same initial output $P_0$, this controller is more highly expressed (larger $\omega_A$) with weaker control in the intra-circuit component and stronger control in the growth-based component (larger $k_\lambda$, larger $k_{B_1}$) (Supplementary Fig. S27, navy). CL$p_A$TL$\lambda$PF similarly enhances performance versus open-loop, but not to the extent of systems (i–iii). However, it does offer an improvement in robustness, with changes in $P_0$, $\tau_{\pm10}$ and $\tau_{50}$ having the standard deviations 6.6% (16.7%), 3.4% (8.6%) and 3.4% (8.6%), with 100% of designs retaining $\tau_{\pm10} = \tau_{90}$ (Supplementary Fig. S24, lime). However, this system requires a very high growth-based control strength (large $k_\lambda$) that may be difficult to achieve in practice (Supplementsry Fig. S27, lime). A comprehensive analysis of the robustness of all multi-input controllers is described in Supplementary Note 3 (Supplementary Figs. S28 and S29 and Supplementary Table S2).

## Negative feedback control increases bioproduction

The design of our negative feedback controllers focused on the maximisation of three primary metrics: $P_0$, $\tau_{\pm10}$ and $\tau_{50}$, with superior systems providing improvements in longevity compared with open-loop systems of the same initial output $P_0$. Such improvements are crucial for applications where reliable performance over time is required, such as logic gates or systems which sense and respond to environmental changes. For bioproduction applications, maximising total production is the primary objective. Noting that some of our controllers exhibit a production/longevity trade-off, we set out to evaluate the ability of the controllers to enhance bioproduction. We proposed two new performance metrics which focus on quantify total protein production from our simulations: $Q$, the cumulative output of protein $p_A$ across the entire evolutionary simulation of repeated batch culture, and $Q_{max}$, the maximum value of $Q$ across all optimal designs. The calculation of these metrics are defined in the Methods section "Cumulative production".

For every system, an increase in initial output $P_0$ corresponds to an increase in cumulative output $Q$, except at very high initial outputs, where the burden is significant enough to reduce total output even where initial output increases (Fig. 9a–b). All controllers, bar CL$p_A$TX, show improvements in production over open loop implementations. This demonstrates that the despite the potential loss in initial output that negative feedback control can create, the enhanced evolutionary stability that control confers enables higher total output over time. In some cases, CL$p_A$TX performs worse than open loop (Fig. 9c, blue). CL$p_A$TL outperforms CL$p_A$TX with the potential to improve on the open-loop cumulative output by up to twofold (Fig. 9c, red). CL$\lambda$TL shows an even greater improvement of up to threefold (Fig. 9c, orange). This suggests that the sRNA-based controllers outperform those based on transcription factors, and that growth-based control outperforms intra-circuit control for this objective. The cumulative outputs of the three multi-input controllers are very similar to each other and very similar to CL$\lambda$TL, suggesting that combining control

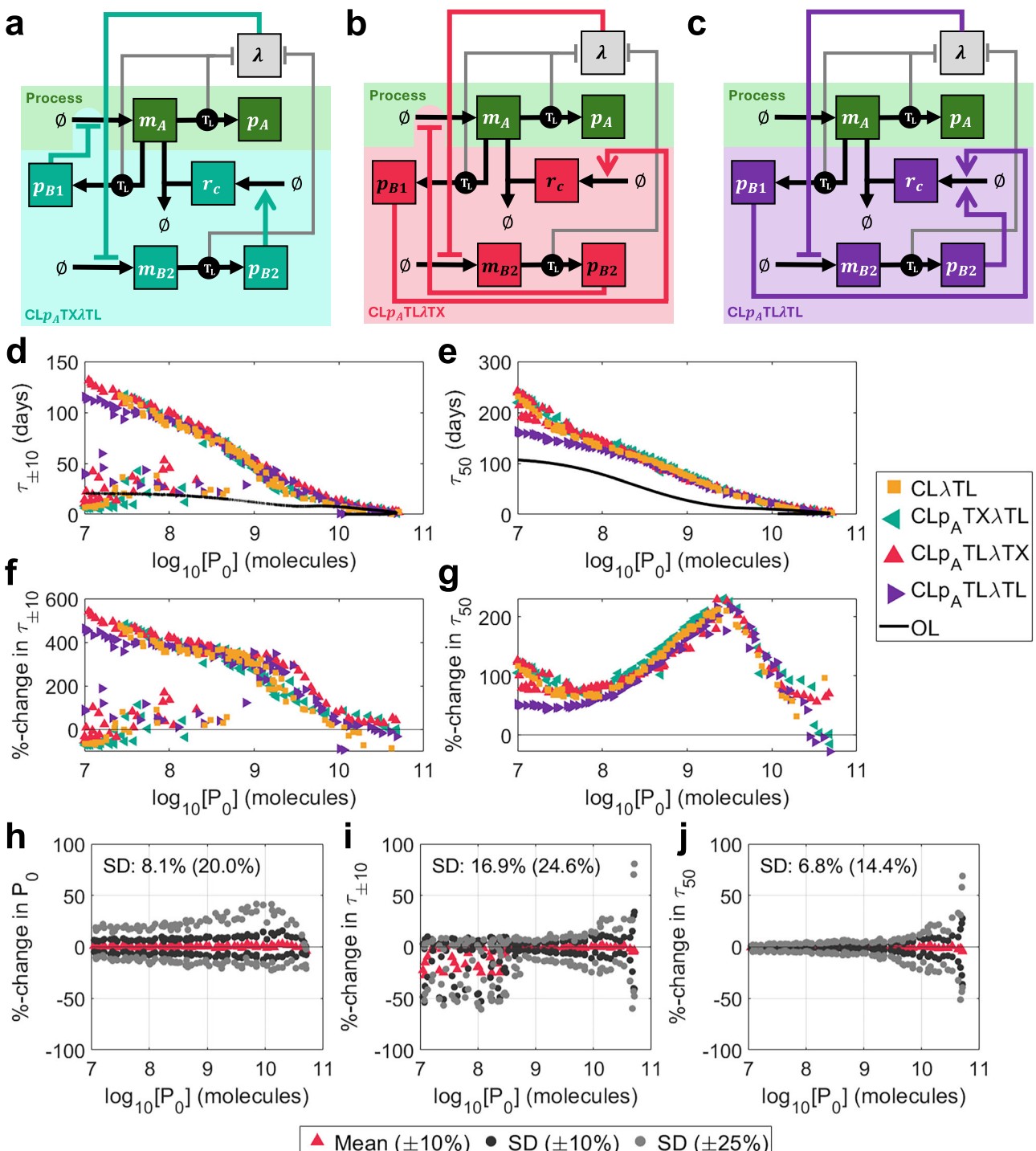

**Fig. 8 | Designing multi-input controllers. a–c** Schematics of mechanistic multi-input controllers: (**a**) CL$p_A$TX$\lambda$TL (protein-based inhibition of transcription, growth-based inhibition of translation, teal), (**b**) CL$p_A$TL$\lambda$TX (protein-based inhibition of translation, growth-based inhibition of transcription, crimson), (**c**) CL$p_A$TL$\lambda$TL (protein-based inhibition of translation, growth-based inhibition of translation, purple). See key in Fig. 2a for symbol meanings. **d–i** Optimised outputs for three multi-input controllers compared with CL$\lambda$TL. (d) $\tau_{\pm10}$ vs initial output $P_0$, (**e**) $\tau_{50}$ vs initial output $P_0$, (**f**) %-change in $\tau_{\pm10}$ over open-loop vs initial output $P_0$, (**g**) %-change in $\tau_{50}$ over open-loop vs initial output $P_0$. **h–j** Robustness analysis for

CL$p_A$TL$\lambda$TX. For each of the 100 optimal controllers with $n_B = 300$ aa, 100 further controllers were generated by varying parameters by up to $\pm10\%$ (dark grey) and $\pm25\%$ (light grey). The percentage changes in three output metrics were calculated versus the original optimal systems: (h) $P_0$, (i) $\tau_{\pm10}$ and (j) $\tau_{50}$. Plots show the means (for $\pm10\%$) and standard deviations (for both $\pm10\%$ and $\pm25\%$) of the percentage changes for each optimal controller. Percentages marked on the plots indicate the standard deviations across the entire Pareto front when parameters were varied by $\pm10\%(\pm25\%)$. Only original optimal controllers where $\tau_{\pm10} = \tau_{90}$ are considered. Simulation results are provided as a Source Data file.

inputs neither harms nor enhances the total cumulative production of a system (Fig. 9b, d).

The maximum cumulative protein production, $Q_{max}$, for each system is as follows: Open-loop, $1.42 \times 10^{11}$ mc; CL$p_A$TX, $1.54 \times 10^{11}$ mc;

CL$p_A$TL, $1.59 \times 10^{11}$ mc; CL$\lambda$TL, $2.47 \times 10^{11}$ mc; CL$p_A$TX$\lambda$TL, $3.13 \times 10^{11}$ mc; CL$p_A$TL$\lambda$TX, $2.46 \times 10^{11}$ mc; CL$p_A$TL$\lambda$TL, $2.44 \times 10^{11}$ mc. Every controller is capable of enhancing the maximum possible cumulative output, even when $Q$ has not been explicitly optimised for, with the best

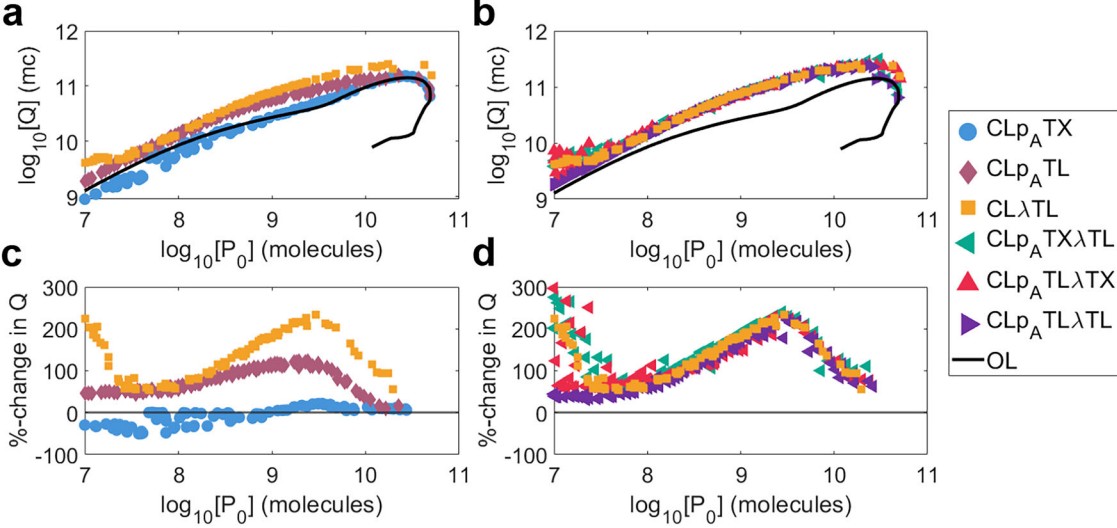

**Fig. 9 | Investigating the cumulative output Q.** We consider an open-loop system (black) and six controllers: (**a**, **c**) CL$p_A$TX (blue), CL$p_A$TL (red), CL$\lambda$TX (orange). **b**, **d** CL$p_A$TX$\lambda$TL (teal), CL$p_A$TL$\lambda$TX (crimson) and CL$p_A$TX$\lambda$TL (purple). **a**, **b** Cumulative output $Q$ against initial output $P_0$. **c**,**d** Percentage-change in $Q$ versus an open-loop system of equal initial output against initial output $P_0$. Each marker represents a controller design from the multi-objective optimisation of $P_0$, $\tau_{\pm10}$ and $\tau_{50}$. Simulation results are provided as a Source Data file.

performing controllers having more than double the maximum cumulative open-loop output. All together these results suggest that the use of evolutionary stabilising negative feedback controllers has the potential to improve yields in industrial bioproduction settings.

## Discussion

Here, we presented designs for feedback controllers which act to stabilise synthetic gene expression over evolutionary time. We developed a 'host-aware' and 'mutation-aware' multi-scale model of microbial gene expression, growth and population dynamics, which mechanistically captures both the interplay between circuits and their host (successfully predicting the growth defect imparted by circuit burden) and the interplay between ancestral cells and their low/non-functional mutants (successfully capturing natural selection)[30–34]. Within this model, we embedded mechanistic models of feedback controllers, implementable as synthetic gene regulatory networks ('circuits') and coupled these host-circuit-controller models with multi-objective optimisations to identify designs which maximise circuit production ($P_0$), maintain the circuit performance close to its ancestral level ($\tau_{\pm10}$) and maintain its presence in the population in the long-term ($\tau_{50}$).

We showed that product-based 'intra-circuit' transcriptional feedback, where a transcription factor is co-transcribed with the circuit gene and inhibits the expression of both proteins, can effectively increase both short-term and long-term performance by increasing $\tau_{\pm10}$ and $\tau_{50}$ compared with an open-loop system of equivalent output $P_0$. Our sensitivity analysis shows a larger design space for increasing $\tau_{\pm10}$ than $\tau_{50}$, suggesting that maintaining performance in the short term is likely easier to achieve in vivo. Our evolutionary simulations show that this strategy acts to maintain protein production at the cellular level in light of mutations in protein production rates. That is, mutant strains with weaker promoters don't have significantly impeded protein outputs. We showed that implementing negative feedback by a transcription factor is effective at nearly doubling $\tau_{\pm10}$ and $\tau_{50}$, but that the additional burden of the controller can abolish any benefit of control; when the regulating protein's length is greater than the *E. coli* average of 300 amino acids, the control scheme can perform worse than open loop in many cases. We then proposed an alternative mechanism for implementing negative feedback, where the process is instead co-transcribed with an activatory transcription factor. This

regulator protein activates the expression of an sRNA which sequesters and silences the mRNA, completing the negative feedback loop. This controller outperforms the transcriptional controller in both the long and short term, with a more than twofold improvement in $\tau_{\pm10}$. Further, this control strategy is less sensitive to controller size, as larger controllers remain capable of enhancing performance versus open-loop. We propose two contributing factors towards this improvement. First, the strong gain of the controller, which enables a smaller quantity of regulator proteins to give rise to a stronger inhibitory action. This is a result of the amplification step created by the transcription of the sRNA, and is the more significant factor in the long term. Second, the potential for mutations to affect the controller gene and yield a small number of slower-growing mutant strains which produce more than the ancestral strain. This factor more significantly affects short-term performance, as the emergence of new mutants makes up a more significant portion of 'mutant spread' early in the time course, before out-competition becomes the predominant factor as a result of growth rate differences. We find that, although designs based on transcriptional feedback have worse optimal performance, they are more robust to parametric uncertainty than those based on sRNA-mediated post-transcriptional feedback, implying that such designs will be easier to engineer in vivo. Our analysis demonstrates that, while sRNA-mediated feedback gives superior performance, this can only be achieved if the parameters can be precisely tuned experimentally. In bacteria, such as E. coli, where metabolic and gene expression resource consumption is dominated by translation, sRNA can be created with little additional cost to the host, enabling a significant negative feedback action for low cost. However, this advantage may not translate to other organisms, such as mammalian cell lines, where recent evidence suggests transcriptional limitations may be significant[40].

Given that growth rate is central in governing the evolutionary longevity of a circuit, and is the key physiological metric impacted by circuit expression, we proposed growth-based feedback as an alternative control strategy for enhancing evolutionary longevity. These controllers inhibit circuit output at low growth. Our evolutionary simulations show that this strategy acts to maintain growth rates in light of mutation, and so mutants do not get the same selective advantage as in the open-loop system. Biologically, these controllers can be implemented using stress-sensitive promoters to drive

expression of a transcription factor[39]. We tested both transcriptional control (where the transcription factor inhibits circuit expression) and post-transcriptional control (where the transcription factor activates an sRNA which silences the circuit mRNA). Both implementations are capable of increasing $\tau_{\pm10}$ and $\tau_{50}$, but, as in the intra-circuit case, the sRNA-mediated feedback mechanism enables greater performance. Further, varying the size of the regulator protein shows that the transcriptional mechanism is much more sensitive to regulator size (i.e., controller burden) than the sRNA-based scheme. The growth-based controllers based on sRNA-mediated feedback achieve similar optimal performance to the intra-circuit feedback in the short term (i.e., similar maximal increases in $\tau_{\pm10}$), but significantly exceed it in the long term, increasing $\tau_{50}$ by 50%. This suggests that growth-based control could be the best strategy for engineering evolutionary longevity, particularly if circuit persistence is of more importance than maintenance of function within a narrow bound. However, our robustness analysis finds that these controllers are much less robust, with some designs falling significantly off the Pareto front. Since population-wide output was the quantity we wanted to maintain over time, we also considered controllers which sense production at the population level, which can be implemented using quorum-sensing. However, these controllers were unable to compete with intra-circuit and growth-based feedback strategies.

To overcome the drawbacks of poor robustness of the growth-based post-transcriptional feedback controller, we developed a number of multi-input controllers which take both protein production and growth as inputs. Whilst our initial resource-free phenomenological models shows such controllers should yield significant improvements in longevity (when compared to single-input topologies), mechanistic models showed that these are sensitive to resource consumption, with additional controller burden abolishing most performance benefits. However, incorporating the additional feedback reduced the sensitivity of a system's output to parametric uncertainty, allowing for enhanced robustness and more reliable performance, without diminishing longevity. These multi-input controllers are therefore the most promising options for in vivo implementation, as they combine the high performance of the growth-based system with the increased robustness that will simplify the selection of biological parts without perfectly tuned parameter values.

Extending our analysis to evaluate the cumulative (total) protein production of our systems over time, we showed that systems with controllers always produce more protein that than open loop (uncontrolled) systems. As we previously found, the best controllers where those based on translational control and those which utilised a growth-based input. Despite not optimising the controller designs to maximise production, we showed that negative feedback could improve total yield despite an initial loss in production per cell. Our results show that increasing evolutionary stability via negative feedback is an attractive strategy for improving yields in industrial bioproduction.

Here, we have conducted a comprehensive analysis of feedback controllers when applied to a process consisting of a single gene over repeated batch cultures. This approach replicates previous experimental investigations of synthetic gene circuit evolution[4,10,20]. Controlling the evolutionary stability of more complex gene circuits represents a more significant challenge, as the dynamics of our negative feedback controllers and the complex processes may interact. It is common practice in control engineering to design a bespoke controller for each individual process. Our controllers can be scaled to stabilise gene expression in systems where gene expression levels are desired to be approximately binary (such as logic gates, activation cascades or induction systems), provided there are sufficient experimental choices of inhibitory regulator (e.g., orthogonal transcription factor proteins or sRNAs) - albeit with controller kinetics adjusted to suit the new process. Whilst controller designs will need to be

optimised for the new processes, we expect our general design rules to hold: that negative feedback will stabilise expression over evolutionary time scales, that translation-based systems will offer superior performance over transcription-based systems (and that the latter may in fact reduce evolutionary stability) and that systems which seek to control function at the population level will be less effective. The choice of intra-circuit or growth-based feedback (and associated differences in design robustness) remains an area for future work. Engineering controllers to stabilise the evolutionary stability of circuits with time-varying dynamics (such as oscillators) represents an additional challenge, given that the introduction of the new controller dynamics may abolish the desired circuit behaviour[41]. Engineering control systems to make such systems more robust and reduce the interactions between host and circuit remains an open question.

The potential impact of synthetic biology is vast, with applications expected to transform the fields of healthcare, chemical production, agriculture and more over the next decade[42]. However, the utility of complex gene circuit designs will be severely limited if they cannot be employed predictably and stably over the timescales required for industrial bioprocesses. It is therefore crucial for circuits to be designed with evolution in mind. While various approaches have been developed to date which improve the longevity of synthetic circuits and pathways, these are often bespoke, rely on selectable markers, or require complex redesign based on the specific process of interest. Here, we have demonstrated that feedback control can be employed effectively to improve evolutionary longevity, both in the short term (i.e., maintaining function close to the designed level) and the long term (i.e., persisting within a population over time). Our designs do not rely on selectable markers (reducing the need for antibiotics at scale) or on conferring a selective disadvantage to mutants (a promising approach, but one which is vulnerable to loss of the toxic module or loss of control of such a module). In this work, we account for controller mutation during our analysis, and therefore our proposed designs are themselves robust to evolutionary change. The proposed approaches can be easily applied to different systems without the need for significant redesign, and can be incorporated alongside existing approaches, contributing towards the ultimate goal of synthetic biology applications that perform reliably in the long term.

## Methods

### Simulating an evolving population of engineered cells

Throughout this paper, we make use of an ODE model capturing the dynamics of an evolving population of engineered cells. This model is similar to one recently developed by Ingram and Stan[34]. The complete circuit-host-population model is described in detail in Supplementary Note 1, alongside complete models of all the control systems considered throughout the paper. Tables of variables and parameters are given in Supplementary Tables S3–S5. Mutation schemes are shown in Supplementary Tables S6 and S7. Simulations are performed which replicate repeated batch conditions. Every 24 h, a representative sample of 1000 cells is selected from the combined population, and the external substrate is reset to $s_X = 10^{12}$ molecules. Over the course of each day, the substrate is consumed and the population grows until the supply is exhausted and the population levels out. This aims to replicate a typical laboratory-based study[10]. This is preferable to chemostat conditions because growth rates are not determined by the chemostat dilution rate, which can cause additional problems if the cells are overburdened and cannot achieve growth fast enough to match it. The model is encoded in MATLAB and we used the inbuilt `ode15s` function to perform the simulations.

### Comparing controllers versus open-loop systems of equal output

To evaluate the performance of a given controller parameterisation $X$ with initial output $P_{0_X}$, we compared its long-term performance ($\tau_{\pm10_X}$

and $\tau_{50_x}$) against an open-loop system of equal initial output. To achieve this, we first generated a large set of 5000 open-loop designs by varying $\omega_A$ over a wide range between 0.1 and 1000 mc min$^{-1}$ (Fig. 2g). This yielded multi-valued relationships between output ($P_0$) and longevity ($\tau_{\pm10}$ and $\tau_{50}$), due to the existence of overburdened systems; when $\omega_A$ is pushed above a certain threshold, both output and longevity fall. To create a set with a single-valued relationship between output and longevity, we removed the overburdened systems. We then used linear interpolation to find the open-loop relationship between output and longevity for a system with an initial output $P_{0_x}$. For figures where we explicitly show the time-series outputs of different systems with comparable initial outputs $P_0$ (e.g., Fig. 3b), we first found the Pareto fronts from a large set of designs which simultaneously maximised $P_0$, $\tau_{\pm10}$ and $\tau_{50}$, then selected points from the fronts with $P_0$ as close as possible to a set value.

## Multi-objective optimisation

Here we outline how we generated optimal designs for the considered controller topologies. For each topology, we first generated a large number of samples using the efast algorithm[43], by varying a large number of parameters over wide biologically feasible ranges. By examining the outputs from the sampling, we identified those parameters **u** which were most significant in determining $P_0$, $\tau_{\pm10}$ and $\tau_{50}$ and used these to perform a multi-objective optimisation to simultaneously maximise these three objectives:

$$\text{maximise}_{\mathbf{u}} \quad (P_0, \tau_{\pm10}, \tau_{50})$$
$$\text{subject to } \mathbf{L} \le \mathbf{u} \le \mathbf{U}. \tag{6}$$

Here, **L** and **U** represent the lower and upper bounds of the parameters **u**. Separate optimisations were performed for each of the considered control topologies, with each being repeated for different controller protein lengths $n_B = 1, 300, 600$ aa (where applicable). The parameters optimised for each controller, and their biologically feasible upper/lower bounds, are defined in Supplementary Table S8. A detailed discussion of the choices of boundary values is given in Supplementary Note 1. To perform the optimisations, we used Matlab's `gamultiobj` genetic algorithm function, with population size 250, Pareto fraction 0.4 and tolerance $10^{-4}$, yielding Pareto fronts consisting of 100 individuals. The fixed and optimised controller parameters are reported in Supplementary Table S8.

## Robustness analysis

Outputs from optimisations yielded Pareto fronts of 100 individuals and a set of optimal parameters **u**. From this set of individuals, we removed any parameterisations with exceedingly poor short-term performance (i.e., where $\tau_{\pm10} \ne \tau_{90}$). For the remaining individuals, we generated 100 alternative parameterisations by randomly varying each parameter by up to $\pm X\%$, for $X = 10, 25$. For example, in the case where $X = 10$, each parameter varied in the optimisation $u_i$ was selected from a uniform distribution with lower bound $0.9u_i$ and upper bound $1.1u_i$, yielding a window of $\pm10\%$ around the original values. In instances where this would push a parameter beyond the upper or lower bounds defined in the multi-objective optimisations, we set the value of the parameter to be exactly the boundary value, to avoid searching a portion of the design space that wasn't available in the optimisations. To evaluate the robustness of individual controllers, we compared these random parameterisations against their original designs, considering the percentage change of five metrics:

1. $P_0$, the initial population-wide output.
2. $\tau_{50}$, the time taken for population-wide output $P$ to halve.
3. $\tau_{\pm10}$, the time taken for population-wide output $P$ to fall outside a window of $\pm10\%$ of its original value.

4. $\tau_{90}$, the time taken for population-wide output $P$ to fall to 90% of its original value.
5. $P_{max}$, the maximum population-wide output $P$ over the course of the simulation.

Standard deviations of these percentages indicate the level of variability in the function of a system at the topological level. In addition, we evaluated the percentage of random designs which showed a discrete change in the value of $\tau_{\pm10}$ due to it no longer being equal to $\tau_{90}$. Throughout the main text, we focus our analysis primarily on $P_0$, $\tau_{\pm10}$ and $\tau_{50}$. A more detailed analysis covering $\tau_{90}$ and $P_{max}$ is provided in Supplementary Note 3. All means and standard deviations for all controllers are presented in Supplementary Table S2.

## Cumulative production

The translation rate $T_{L_A}$ of protein $p_A$ gives the rate of production of output protein $p_A$ per cell. Integrating $T_{L_A}$ over the time course of the simulation, therefore, gives the total production of $p_A$ per cell. We define the cumulative output $Q$ of a system to be the total production of $p_A$ across an entire culture over the course of a simulation. For a system composed of $n$ mutation states, with population sizes $N_i$ and translation rates $T_{L_{A_i}}$, $Q$ is defined as follows:

$$Q = \sum_{i=1}^{n} \int_{t=0}^{t_{\text{end}}} T_{L_{A_i}} N_i \, dt.$$

We define $t_{\text{end}}$ to be the point where output falls below 1% of its initial value $P_0$. We calculated the value of $Q$ for each controller design given by the multi-objective optimisations. Across those optimal designs, we designate the maximum value of $Q$ as $Q_{max}$. To assess the ability of a given controller architecture to improve total bioproduction, we compared $Q_{max}$ against the maximum cumulative production for an open-loop system. Note that $Q_{max}$ and $Q_{max_{OL}}$ don't necessarily correspond to the same initial output $P_0$.

## Reporting summary

Further information on research design is available in the Nature Portfolio Reporting Summary linked to this article.

## Data availability

Optimisation results generated from this study and are available in the Source Data file. Source data are provided in this paper.

## Code availability

The code used to develop the model, perform the analyses and generate results in this study is publicly available and has been dpositite on Github at https://github.com/apsduk/byrom-nat-commun-2025 under a CC-BY 4.0 license. The specific version of the code associated with this publication is archived in Zenodo and citable with https://doi.org/10.5281/zenodo.16094166[44].

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

## Acknowledgements

D.P.B. is supported by an EPSRC DTP studentship (EP/W524645/1, project reference 2686012). A.P.S.D. is supported by a Royal Academy of Engineering Research Fellowship (RF/202021/20/270) and acknowledges funding from EPSRC (EP/Y00342X/1, EP/X039587/1) and BBSRC (BB/Y007603/1, BB/Z517124/1).

## Author contributions

A.P.S.D. conceived and designed the research. D.P.B. developed the models, conducted all computations, and plotted the results. D.P.B. and A.P.S.D. discussed the results and wrote the paper.

## Competing interests

The authors declare no competing interests.
