## [Transparent Peer Review file · Nature Communications]

Genetic controllers for enhancing the evolutionary longevity of synthetic gene circuits in bacteria

Corresponding Author: Dr Alexander Darlington

Version 0:

Reviewer comments:

Reviewer #1

(Remarks to the Author)

In this work, Byrom et al. proposed several control strategies and evaluated their effectiveness in enhancing the long-term stability of synthetic biological systems. Their modeling framework builds on their previous studies, incorporating the interplay between host and circuit gene expression, circuit mutation, and competition between mutants. The study compares three levels of negative feedback: intracircuit, growth-based, and population-based control. Notably, they found that population-based control was the least effective. They further analyzed transcriptional and post-transcriptional implementations of both intracircuit and growth-based controllers, as well as their combinations. Overall, this work provides a comprehensive and systematic in silico analysis, with a well-designed simulation framework and solid comparisons.

Below are some minor comments.

1. The authors appropriately scaled the transcriptional rate of the circuit gene to ensure comparable protein levels across all cases, maintaining a fair comparison. However, the exact implementation remains unclear. Equation 2 needs clarification—was w_A rescaled in the controller system, the no-controller case, or both? Additionally, how was the same protein level ensured? The authors should provide a justification and, ideally, demonstrate this in a figure (e.g., Fig. 3b) by showing the protein level before mutations were introduced.
2. Some figure legends need clarification. For instance, what does the grey curve represent in Fig. 2b? Also, in addition to a box, text annotations should be added for the Fig. 2d-e legend.
3. The introduction of Fig. 4b should come earlier to provide context before discussing the results in Fig. 4c-h.
4. It would be beneficial to include standard deviation plots for the output metrics under parameter variation, particularly comparing CLpATX and CLpATL.
5. Why was the combination case CLpATX λ TX not considered in the final section? Providing a rationale for this omission would strengthen the analysis.
6. The Methods section mentions five performance metrics, but the results for τ_{90} and P_{max} across different controller designs are not explicitly discussed. Including this analysis would enhance completeness.
7. The authors could consider citing Ref. PMID: 38320912 to provide additional context on the burden imposed by circuit expression if possible.
8. Are there any insights or justifications for the nominal parameter selection and the range used for optimization? A brief discussion on this would improve transparency.
9. It is strongly recommended that the provided code be shared in a public repository (e.g., GitHub) to facilitate reproducibility and further research.

(Remarks on code availability)

Reviewer #2

(Remarks to the Author)

Summary

In this paper, Byrom and Darlington systematically analyze different feedback control topologies for the ability to make gene expression robust to burden-driven evolutionary pressures. They find that different controller designs optimize for different metrics of evolutionary stability, with controllers operating at the translational level generally performing better and with

differences in outcomes depending on whether controllers sense the output protein in individual cells vs the population or instead sense growth rate changes directly. Through their analyses and optimizations, they identify that mutations to controllers and their controlled output genes can offset to balance in some cases, contributing to short-term gene expression stability. The study of mutation accumulation and effects on both the controller and output genes extends prior work on feedback controllers for mutational burden. More broadly, the manuscript sets a standard for host-aware modeling and analysis of protein production in microbes by incorporating the contributions of energy (abstracted metabolism), ribosome loading, and genetic mutation to cell growth and protein production. The analysis very thorough, is well-considered with respect to in vivo implementations, and will accelerate efforts to engineer more robust gene circuits across diverse organisms and for diverse applications, particularly in bioproduction.

I believe that this paper will be of great interest to the synthetic biology community and Nature Communications readership, and is suitable for publication following minor revisions as per below.

Major Notes

- In the context of protein production, the desire to maximize overall product output may be at odds with reductions in output by feedback control. Is improved longevity able to make up for lower starting levels of the output? The manuscript compares the CL designs to OL systems with similarly-reduced outputs, which makes sense for showing that the controllers increase longevity and robustness better than simply reducing output. However, it would still be important for max bioproduction applications to show the cumulative production over time for a system +/- controller, but with other parameters unchanged.
- The interpretation that CLpATL outperforms CLpATX depends in part on the assumption of no burden at the transcriptional level. While it's been shown that translational burden by ribosome loading is dominant in bacteria, transcriptional burden is still important to consider in these organisms. In addition, transcriptional burden is more dominant in eukaryotes, which will be important to consider for generalizing the results here to diverse systems. Some discussion of the effects of transcriptional burden would thus be warranted, minimally in the text where appropriate, but ideally also through a model of a subset of controllers that accounts for both TX and TL resources and assigns relative weights to each to determine how burden at each level affects OL vs CL circuit performance metrics. This analysis would improve the broad application of this manuscript.

Minor Notes

Results

- Figures S1 and S3-9 should be referenced in the main text where relevant (for those connected to Supp Notes, in the same place as referring to the Notes)
 - I found S1 to be very informative – it could be incorporated into Fig 1
- Page 11: “As controller burden increases, the parametric design rules do not qualitatively change” -- some different behavior is observed in the KB plot as nB increases from 1 to 300/600, where it looks like the trend is inverted. Does this reflect different optimal KB values as a function of # AAs?
- Page 13: Reference to Fig. S2 at the end of the first paragraph says “Fig 2”
- Page 15, last paragraph: CLATL outperforming CLATX is true for most cases except at high P0 levels for the T50 metric (per Fig S10f), which is important to point out here
- For the combined growth rate + intra-circuit controllers that both operate via sRNA, would it be possible to combine the sensors into a single promoter/gene, and would that improve the performance?

Discussion

- Page 22, line 2: “without penalising output P0” is not exactly true, since the negative feedback necessarily reduces the output level, but is true for a given set point level compared to OL equivalents.
- Can the authors comment on scalability of the controller designs for systems with increasing numbers of genes to control, and for situations where dynamic gene regulation may be involved?

Figures

- Fig 2c-e: Include a descriptor in the legend about what the different colors mean. They can be inferred from Fig. 1 but this will improve clarity.
- Fig 3c-d: Include descriptor of what “x” means in the plots – I’m assuming it corresponds to the single timecourse in panel (b).
- Fig 3e-f, S20e-f: Shading the colored and outlined boxes differently for each mutational state would help to connect the information in (e) to that in (f). Another option would be combining into a single plot of pA vs λ (in which case the T0 vs T50 boxes would be different shades)
- Fig 4f-g shows reduced T10 and T50 at the highest levels of P0 – does this imply that the controllers are decreasing robustness of output levels in that regime?
- Fig 4i-j, 6h-i, 8j-o, S12, S15, S18: I think it would be more clear if the shading of +/-10% was dark grey (for uniformity) or a darker shade of the variant color itself, rather than using the dark red shade associated w/ CLpATL
- Fig 4i-l: not explicitly referenced in the main text (presumably should be on Page 11, paragraph 2).
 - It's also relatively unclear how to interpret beyond what is said in the main text, since the graphs do not directly show SD/CV, but rather the raw values. The graphs here and comparable ones elsewhere (e.g. Fig 6hi, Fig 8j-o, S12, S15, S18)

could be improved by adding an inset measurement of SD/CV.

o If space/format permits, consider moving Supp Table 1 to main text – it's very helpful as a summary of this robustness point

• Fig 5: Indicate somewhere that $mc :=$ molecules, for clarity

Supplement

• Equation 28 (and other equivalents w/ multiple proteins from one mRNA, like eq 34):

o As-written, it appears that the translation of one ORF (i.e. proteins A vs B) affects the translation of the other due to mutually-exclusive formations of complexes w/ ribosome (CA, CB). This might not significantly affect the model, but translation of each ORF should be decoupled, as the ribosome can bind to each RBS independently.

• Equation 34:

o Should the $\Phi(\rho_B)$ be there in the TXA term?

o Here and in several other eqs (including 39, 50, 56, and more), it appears that the species rC should be mC ?

(Remarks on code availability)

READ_ME file is included and clear, code is clear on quick examination. I did not run the code to test it, though (do not currently have MATLAB license).

Version 1:

Reviewer comments:

Reviewer #1

(Remarks to the Author)

The authors have fully addressed all my comments.

(Remarks on code availability)

Reviewer #2

(Remarks to the Author)

Thank you to the authors for your revisions and consideration of all reviewer comments. As with the rest of the manuscript, the responses and revisions were very detailed, well-considered, and high-quality. I would recommend publication, with very minor typographic revision of the following:

Fig S26 is referenced in main text prior to S24-5, so consider re-arranging these figures in the SI

Refs for most supplementary tables in main text are missing

(Remarks on code availability)

No additional comments

Response to Reviewers

Genetic controllers for enhancing the evolutionary longevity of synthetic gene circuits in bacteria

Daniel P. Byrom¹ and Alexander P.S. Darlington^{1,*}

¹Warwick Integrative Synthetic Biology Centre, School of Engineering, University of Warwick, Coventry, CV4 7AL, UK

* correspondence to a.darlington.1@warwick.ac.uk

Reviewer 1

In this work, Byrom et al. proposed several control strategies and evaluated their effectiveness in enhancing the long-term stability of synthetic biological systems. Their modeling framework builds on their previous studies, incorporating the interplay between host and circuit gene expression, circuit mutation, and competition between mutants. The study compares three levels of negative feedback: intracircuit, growth-based, and population-based control. Notably, they found that population-based control was the least effective. They further analyzed transcriptional and post-transcriptional implementations of both intracircuit and growth-based controllers, as well as their combinations. Overall, this work provides a comprehensive and systematic in silico analysis, with a well-designed simulation framework and solid comparisons. Below are some minor comments.

1. The authors appropriately scaled the transcriptional rate of the circuit gene to ensure comparable protein levels across all cases, maintaining a fair comparison. However, the exact implementation remains unclear. Equation 2 needs clarification — was w_A rescaled in the controller system, the no-controller case, or both? Additionally, how was the same protein level ensured? The authors should provide a justification and, ideally, demonstrate this in a figure (e.g., Fig. 3b) by showing the protein level before mutations were introduced.

We thank the reviewer for highlighting where our manuscript requires clarification. We have made a number of changes to improve the communication of our results:

- We have added a new supplementary Figure S2 which contains time-series plots similar to Fig. 3b to demonstrate how reducing the open-loop transcription rate ω_A or applying control to a fixed process through any mechanism will improve longevity at the expense of output. This makes it more clear what happens when ω_A is not tuned in the open-loop setting to create similar protein levels.
- We have made reference to Figure S2 in the main text on page 7, noting the open-loop result:

“It is trivial to reduce burden and improve both $\tau_{\pm 10}$ and τ_{50} by reducing the production of p_A via the birth rate ω_A . However, this impedes circuit function (Fig. S2a).” and on page 9, noting the closed-loop result “For all three systems, applying control to a given nominal process (here, $\omega_A = 10 \text{ mc min}^{-1}$) boosts longevity at the expense of initial output P_0 , with stronger controllers (low k_A , high k_λ , low k_P) causing a more significant change (Fig. S2b-d).”

- We have added a new section to the methods (Methods 4.2) to describe more clearly how we compared a closed-loop design with specific output P_0 against an open-loop design of equal initial output P_0 : “To evaluate the performance of a given controller parameterisation X with initial output P_{0X} , we compared its long-term performance ($\tau_{\pm 10X}$ and τ_{50X}) against an open-loop system of equal initial output. To achieve this, we first generated a large set of 5000 open-loop designs by varying ω_A over a wide range between 0.1 and 1000 mc min^{-1} (Fig. 2g). This yielded multi-valued relationships between output (P_0) and longevity ($\tau_{\pm 10}$ and τ_{50}), due to the existence of overburdened systems; when ω_A is pushed above a certain threshold, both output and longevity fall. To create a set with a single-valued relationship between output and longevity, we removed the overburdened systems. We then used linear interpolation to find the open-loop relationship between output and longevity for a system with initial output P_{0X} . For figures where we explicitly show the time-series outputs of different systems with comparable initial outputs P_0 (e.g. Fig. 3b), we first found the Pareto fronts from a large set of designs which simultaneously maximised P_0 , $\tau_{\pm 10}$ and τ_{50} , then selected points from the fronts with P_0 as close as possible to a set value.” This is referenced on page 7 at the end of section 2.1: “See Methods 4.2 for full details as to how these metrics were determined.”
- To clarify our use of language, we can confirm that P_0 is defined as the “protein level before mutations were introduced” and at this point we assume the system does not contain mutants. Running the simulation without mutation simply leads to horizontal straight lines at P_0 as in Fig. 2c.
- We have updated the captions of Fig. 3 and Fig. S23 to explicitly state the parameters used to yield systems of equal initial output, and described how these were chosen. Briefly, we generated a large number of designs for both open-loop and closed-loop systems and presented the time-series outputs for those systems with initial output P_0 closest to a fixed value (in this case $2 \times 10^9 \text{ mc}$). We have modified the captions for Fig. 3b-d as follows “(b) Time-series of population-wide output P over time for an open-loop system (black, dashed, $w_A = 4.0 \text{ mc min}^{-1}$) and representative control systems of equivalent initial output P_0 . (Intra-circuit: red, $w_A = 10^3 \text{ mc min}^{-1}$, $k_A = 4.5 \times 10^2$

mc. Growth-based: orange, $w_A = 87 \text{ mc min}^{-1}$, $k_\lambda = 6.3 \times 10^{-2} \text{ min}^{-1}$. Population-based: lilac, $w_A = 16 \text{ mc min}^{-1}$, $k_P = 5.2 \times 10^8 \text{ mc}$). (c-d) A large number of designs were generated by varying the maximal transcription rate ω_A and control parameters k_u . Against the initial output P_0 , we plot the percentage change in (c) $\tau_{\pm 10}$ and (d) τ_{50} vs an open-loop system of equal initial output. Points marked with an X correspond to the time-series plots in (b). These were selected as points on the Pareto fronts simultaneously optimising P_0 , τ_{50} and $\tau_{\pm 10}$, with initial output P_0 closest to 2×10^9 molecules.” We have updated the figure caption for the multi-input version of this figure (Fig. S23) in an identical manner.

2. Some figure legends need clarification. For instance, what does the grey curve represent in Fig. 2b? Also, in addition to a box, text annotations should be added for the Fig. 2d-e legend.

We have reviewed and amended the figure legends throughout the paper. For Fig. 2c (previously Fig. 2b), we have added a legend indicating the definition of the grey line. We have added a new panel Fig. 2b which serves as a legend for the mutation states in Fig. 2, as well as presenting the possible transitions between them. For Fig. 2e-f (previously panels d-e) (and correspondingly Fig. 7d-e), we have removed the inset boxes, replacing them with horizontal lines which indicate the maximum production or maximum growth rate over a single day. The revised caption for Fig. 2 is given as follows:

“Simulating an open-loop process in repeated batch conditions. (a) A schematic showing the function of the process. m_A is spawned and translated to create output p_A using host ribosomes. This process impacts host growth. The total output P comes from the output p_A across an entire population of engineered cells. (b) A visual depiction of the mutation scheme for this process. Coloured squares represent distinct ‘mutation states’ with different levels of function. Numbers in the squares show the percentage function of a given state relative to the designed level, implemented through differences in the maximal transcription rate ω_A : 100 represents a state functioning as designed, while 0 represents a completely non-functional state where no transcripts m_A are produced. Arrows signify possible transitions between mutation states, with lighter arrows representing mutations which occur less often. (c-f) Time-series outputs using an open-loop process with maximal transcription rate $\omega_A = 5 \text{ mc min}^{-1}$. (c) Total population-wide circuit output P , plotted both in full (grey) and at the end of each simulation day (green). An ideal system would match an open-loop system in the absence of mutation, maintaining function indefinitely (blue). (d) Population size N , distributed according to mutation state. Dark green represents a fully-functional (100%) strain. Light green represents a non-functional (0%) strain. (e) Output per cell p_A according to mutation state over the first day. Dotted lines show maximum

outputs. (f) Growth rate λ according to mutation state over the first day. Dotted lines show maximum growth rates. (g) For a wide range of processes (varying ω_A between 0.1 and 1000 mc min^{-1}), $\tau_{\pm 10}$ (black) and τ_{50} (grey) against initial output P_0 .”

3. The introduction of Fig. 4b should come earlier to provide context before discussing the results in Fig. 4c-h.

To improve the flow of the text, we have amended the main text to introduce both the $CLp_A TX$ (Fig. 4a) and the $CLp_A TL$ controllers (Fig 4b) before discussing their respective performances. The first paragraph of Section 2.3 has been amended:

“Here we consider two biologically implementable control systems where feedback is enacted in proportion to the per-cell protein output p_A . The first is composed of an inhibitory transcription factor p_B which is produced from the same promoter as circuit protein p_A , so that the functional identity of p_A is not constrained.”

We have made further minor wording changes to this section to improve grammar given the above change.

4. It would be beneficial to include standard deviation plots for the output metrics under parameter variation, particularly comparing $CLp_A TX$ and $CLp_A TL$.

In light of this comment and that of Reviewer 2 (comment 17) we have included the standard deviation plots as requested and updated our presentation of the robustness results for all controllers as follows:

- We have calculated standard deviations for the percentage change of P_0 , $\tau_{\pm 10}$ and τ_{50} versus the corresponding original parameterisation and replaced the original ‘front figures’ with these - the original figures are now available as supplementary figures.
- For Figure 4 (the $CLp_A TX$ and $CLp_A TL$ results), these results are shown as new panels (g-l). We have amended the caption of Fig. 4 to include the following: “(g-l) Robustness analyses for (g-i) $CLp_A TX$ and (j-l) $CLp_A TL$. For each of the 100 optimal controllers with $n_B = 300$ aa, 100 further controllers were generated by varying parameters by up to $\pm 10\%$ (dark grey) and $\pm 25\%$ (light grey). The percentage changes in three output metrics were calculated versus the original optimal systems: (g,j) P_0 , (h,k) $\tau_{\pm 10}$ and (i,l) τ_{50} . Plots show the means (for $\pm 10\%$) and standard deviations (for both $\pm 10\%$ and $\pm 25\%$) of the percentage changes for each optimal controller. Percentages marked on the plots indicate the standard deviations across the entire Pareto front when parameters were varied by $\pm 10\%$ ($\pm 25\%$).”
- We have repeated this change for all other controllers throughout the paper (Fig. 6f-h, 8h-j, S3h-j,

S11g-i, S16h-j, S21h-j, S26i-n).

- We have created new supplementary figures which give standard deviation plots for the additional metrics P_{\max} and τ_{90} as well as the original robustness plots showing the absolute values (Fig. S5, S8, S19, S25, S28, S29).
- We have written a comprehensive description of the robustness of all of the the controllers under study and present this in new Supplementary Note SN3.

5. Why was the combination case $CLp_A TX\lambda TX$ not considered in the final section? Providing a rationale for this omission would strengthen the analysis.

Based on our initial analysis that sRNA-mediated feedback outperforms transcriptional feedback, we assumed this controller would perform worse than those already present in the original submission. To validate our assumption, we have now developed a model of the proposed $CLp_A TX\lambda TX$ system and conducted a performance and robustness analysis with results shown in Fig. S26 (controller schematic, optimised performance and primary robustness metrics), Fig. S27 (optimal designs, i.e. parameters) and Fig. S29 (the additional robustness metrics). We find, as expected, this controller performs worse than those in our original submission. We now introduce this controller in the main text as an additional example of a multi-input control system “(iv) $CLp_A TX\lambda TX$ uses both protein-based feedback and growth-based feedback to co-operatively inhibit transcription of the process gene.” Based on a comment of our second reviewer (Reviewer 2, comment 8), we have also incorporated a fifth multi-input controller $CLp_A TL\lambda PF$. We present the results of these two new multi input controllers in the main text as follows:

“Although $CLp_A TX\lambda TX$ enhances performance versus open-loop, it performs worse than systems (i)-(iii) without providing a notable improvement in robustness: changes in P_0 , $\tau_{\pm 10}$ and τ_{50} have the standard deviations 11.1% (28.1%), 19.1% (28.6%) and 4.2% (10.1%) (Fig. S26, navy). 80.6% of controllers retain $\tau_{\pm 10} = \tau_{90}$. At the same initial output P_0 , this controller is more highly expressed (larger ω_A) with weaker control in the intra-circuit component and stronger control in the growth-based component (larger k_λ , larger k_{B1}) (Fig. S27, navy). $CLp_A TL\lambda PF$ similarly enhances performance versus open-loop, but not to the extent of systems (i)-(iii). However, it does offer an improvement in robustness, with changes in P_0 , $\tau_{\pm 10}$ and τ_{50} having the standard deviations 6.6% (16.7%), 3.4% (8.6%) and 3.4% (8.6%), with 100% of designs retaining $\tau_{\pm 10} = \tau_{90}$ (Fig. S26, lime). However, this system requires a very high growth-based control strength (large k_λ) that may be difficult to achieve in practice (Fig. S27, lime). A comprehensive analysis of

the robustness of all multi-input controllers is described in Supplementary Note 3 (Fig. S28-S29, Table S1).”

6. The Methods section mentions five performance metrics, but the results for τ_{90} and P_{\max} across different controller designs are not explicitly discussed. Including this analysis would enhance completeness.

We have added a comprehensive robustness analysis in new Supplementary Note SN3 where we explicitly discuss these additional metrics in detail. We have amended Methods 4.4 to note this:

“Throughout the main text, we focus our analysis primarily on P_0 , $\tau_{\pm 10}$ and τ_{50} . A more detailed analysis covering τ_{90} and P_{\max} is provided in Supplementary Note SN3.”

Further, in every case where robustness metrics are presented in the main text, this additional analysis is referenced, e.g. as follows for *CLp_ATX*: “See Supplementary Note SN3 for our comprehensive robustness analysis (Fig. S8a-d, Table S1).” Standard deviations of these metrics are plotted for each controller (Fig. S5, S8, S19, S28, S29).

7. The authors could consider citing Ref. PMID: 38320912 to provide additional context on the burden imposed by circuit expression if possible.

We have added the reference (as reference [9]) to the introduction as follows:

“This additional load imparted by the circuit often leads to a gross reduction in the cell growth rate, a phenomenon known as “burden” (reviewed extensively in [6, 7, 8, 9]).”

8. Are there any insights or justifications for the nominal parameter selection and the range used for optimization? A brief discussion on this would improve transparency.

The host model parameterisation is taken from established models and is referenced in Supplementary Tables S3, S4 and S8. To improve transparency of our multiobjective optimisation approach we have updated Supplementary Note SN1 to include a new section SN1.6 which explicitly details our rationale, with references, for the choices of upper and lower bounds for each parameter varied. This supplementary note is referenced in Methods 4.3, where the multi-objective optimisation process is defined:

“The parameters optimised for each controller, and their biologically feasible upper/lower bounds, are defined in Table S8. A detailed discussion of the choices of boundary values is given in Supplementary Note SN1.6.”

9. It is strongly recommended that the provided code be shared in a public repository (e.g., GitHub) to facilitate reproducibility and further research.

The code will be made available online upon acceptance. We have updated the Code Availability section of the paper:

“All computational models and analyses were developed in MATLAB 2022b. The authors declare that MATLAB codes of the models and optimisations, for replicating the results presented in the main body, are available at GitHub <https://github.com/apsduk/byrom-nat-commun-202#>. The code repository is also deposited in Zenodo and citable with doi [10.5281/zenodo.#####](https://doi.org/10.5281/zenodo.#####).”

Reviewer 2

Summary

In this paper, Byrom and Darlington systematically analyze different feedback control topologies for the ability to make gene expression robust to burden-driven evolutionary pressures. They find that different controller designs optimize for different metrics of evolutionary stability, with controllers operating at the translational level generally performing better and with differences in outcomes depending on whether controllers sense the output protein in individual cells vs the population or instead sense growth rate changes directly. Through their analyses and optimizations, they identify that mutations to controllers and their controlled output genes can offset to balance in some cases, contributing to short-term gene expression stability. The study of mutation accumulation and effects on both the controller and output genes extends prior work on feedback controllers for mutational burden. More broadly, the manuscript sets a standard for host-aware modeling and analysis of protein production in microbes by incorporating the contributions of energy (abstracted metabolism), ribosome loading, and genetic mutation to cell growth and protein production. The analysis very thorough, is well-considered with respect to in vivo implementations, and will accelerate efforts to engineer more robust gene circuits across diverse organisms and for diverse applications, particularly in bioproduction.

I believe that this paper will be of great interest to the synthetic biology community and Nature Communications readership, and is suitable for publication following minor revisions as per below.

Major Notes

1. In the context of protein production, the desire to maximize overall product output may be at odds with reductions in output by feedback control. Is improved longevity able to make up for lower starting levels of the output? The manuscript compares the CL designs to OL systems with similarly-reduced outputs, which makes sense for showing that the controllers increase longevity and robustness better than simply reducing output. However, it would still be important for max bioproduction applications

to show the cumulative production over time for a system +/- controller, but with other parameters unchanged.

We recognise that our initial analysis did not take into account cumulative production, which would be useful for bioproduction applications. We have extended the main text to consider an additional analysis based on the total cumulative output over the course of a simulation Q . This is presented in the new results section, Section 2.7, and corresponds to the new figure Fig. 9. As with $\tau_{\pm 10}$ and τ_{50} , we compare the value of Q against an open-loop system of equal output P_0 . However, we also consider the metric Q_{\max} , giving the maximum cumulative production across all designs, to consider the potential of feedback for maximum bioproduction. Our results coincide with those earlier in the paper with translation-based feedback capable of producing more total output than transcription-based feedback and growth-based feedback producing more than intra-circuit feedback. While combining control inputs doesn't provide an additional benefit over simple growth-based control at the same P_0 , it does provide larger values of Q_{\max} . We have also incorporated a new Methods section 4.5 to define the metrics Q and Q_{\max} and detail our analysis. Our results are summarised in a new paragraph of the discussion as follows:

“Extending our analysis to evaluate the cumulative (total) protein production of our systems over time, we showed that systems with controllers always produce more protein than open loop (uncontrolled) systems. As we previously found, the best controllers were those based on translational control and those which utilised a growth-based input. Despite not optimising the controller designs to maximise production, we showed that negative feedback could improve total yield despite an initial loss in production per cell. Our results show that increasing evolutionary stability via negative feedback is an attractive strategy for improving yields in industrial bioproduction.”

2. The interpretation that CLpATL outperforms CLpATX depends in part on the assumption of no burden at the transcriptional level. While it's been shown that translational burden by ribosome loading is dominant in bacteria, transcriptional burden is still important to consider in these organisms. In addition, transcriptional burden is more dominant in eukaryotes, which will be important to consider for generalizing the results here to diverse systems. Some discussion of the effects of transcriptional burden would thus be warranted, minimally in the text where appropriate, but ideally also through a model of a subset of controllers that accounts for both TX and TL resources and assigns relative weights to each to determine how burden at each level affects OL vs CL circuit performance metrics. This analysis would improve the broad application of this manuscript.

We agree that in some settings transcriptional burden can become dominating and therefore we extended our models of the $CLp_A TX$ and $CLp_A TL$ to include resource utilisation by transcription. We find that unless transcription is highly burdensome $CLp_A TL$ continues to outperform $CLp_A TX$. We have included a new Supplementary Note SN6 and Supplementary Figure S15 and described the new results in the main text at the end of section 2.4 on page 15:

“We have shown that $CLp_A TL$ outperforms $CLp_A TX$ partly as a result of the low burden of transcription compared with translation. We extended our cell model to capture energy consumption, and therefore burden, by transcription. We found that $CLp_A TX$ is insensitive to this additional burden and that $CLp_A TL$ outperforms it except at the highest transcriptional burden (Fig. S15). See Supplementary Note SN6 for a comprehensive discussion. Given this observation, we proceed with the assumption that transcriptional burden remains negligible and that translational demands (in terms of both energy consumption and cellular resource competition) dominate bacterial growth and host-circuit interactions, in accordance with established models and experimental data [31, 19, 36, 37, 38].”

We have also proposed to update the title of our paper to include the words “*in bacteria*” to ensure that readers are clear as to the wider applicability of our designs. Finally, we have amended the discussion of our manuscript as follows:

“In bacteria, such as *E. coli*, where metabolic and gene expression resource consumption is dominated by translation, sRNA can be created with little additional cost to the host enabling a significant negative feedback action for low cost. However, this advantage may not translate to other organisms, such as mammalian cell lines, where recent evidence suggests transcriptional limitations may be significant [40].”

Minor Notes

Results

3. Figures S1 and S3-9 should be referenced in the main text where relevant (for those connected to Supplementary Notes, in the same place as referring to the Notes)

We have reviewed the text of the paper to ensure that all Supplementary Figures are referenced in order in the main text, and that Supplementary Notes are referred to in order.

4. I found S1 to be very informative – it could be incorporated into Fig 1

We have added an additional panel to new Figure 2 (now Fig. 2b) which shows the 1D mutation scheme for a system with a single synthetic gene. This is a simplified version of Fig. S1, which we hope is

clearer to the uninitiated reader than Fig. S1 (which contains additional dimensions of mutation not relevant until Section 2.3). This panel is referenced on page 5:

“Mutation occurs via transition rates between these populations such that only function-reducing mutations may occur, and such that more extreme mutations are less likely (Fig. 2b). (See Supplementary Note 1.4 and Fig. S1 for a full description of the mutation scheme.)”

At later parts in the paper, when the 2D and 3D mutation schemes are introduced, we’ve explicitly referenced the more complex supplementary figure, e.g. on page 11:

“We assume that mutations only affect a single promoter at a time (Fig. S1).” to help the reader.

5. Page 11: “As controller burden increases, the parametric design rules do not qualitatively change” – some different behavior is observed in the KB plot as n_B increases from 1 to 300/600, where it looks like the trend is inverted. Does this reflect different optimal KB values as a function of # AAs?

Our original comment that qualitative design rules do *not* change was based on comparisons of $n_B = 300$ and $n_B = 600$ amino acids - i.e. if the regulator protein doubles in size we do not see new trends in design rules. The reviewer is correct that changes are apparent when one moves to an effective ‘no burden’ regulator (i.e. when the regulator length is 1 amino acid, $n_B = 1$). We have now made changes to the main text to provide more detail about the optimal parameters and account more clearly for our observation. Firstly, we have amended the initial discussion of the the parameters for $CLp_A TX$ on page 11:

“Our optimisations show that, where improvements are possible, control strength should be maximised (i.e. low k_B) across the Pareto front (Fig. S7, blue). This is because stronger inhibitors do not need to be as abundant to achieve the same strength of feedback and so controller burden can be minimised. At low outputs P_0 , where performance increases are minimal or non-existent, controller binding strength b_B is very small, so there is less pressure to minimise k_B as controller proteins are less abundant.”

We have also amended the discussion for $n_B = 1$ aa (page 12):

“The binding rate b_B can be increased without significantly increasing burden (Fig. S7, blue).”

We have also amended the first paragraph of Section 2.4 (page 12) for $CLp_A TL$ to reflect a similar observation:

“For $n_B = 300$ aa and $n_B = 600$ aa, the optimisation is highly incentivised to find controllers

with controller proteins as few and as strong as possible, to get “more control for less burden”. For $n_B = 1$ aa, there is much less controller burden, and therefore not so much incentive to completely minimise k_B .”

6. Page 13: Reference to Fig. S2 at the end of the first paragraph says “Fig 2”

We have reviewed the text and corrected typographical errors and cross references, including this one. We thank the reviewer to highlighting the error.

7. Page 15, last paragraph: $CL\lambda TL$ outperforming $CL\lambda TX$ is true for most cases except at high P_0 levels for the T50 metric (per Fig S10f), which is important to point out here

We have amended the text clarify this point:

“Both $CL\lambda TX$ and $CL\lambda TL$ improve evolutionary performance, with $CL\lambda TL$ outperforming $CL\lambda TX$ in both the short term (greater $\tau_{\pm 10}$) and long term (greater τ_{50} except at very high initial outputs P_0) (Fig. S16).”

8. For the combined growth rate + intra-circuit controllers that both operate via sRNA, would it be possible to combine the sensors into a single promoter/gene, and would that improve the performance?

We propose one biological mechanism to implement this kind of control scheme would be to place the sRNA inhibitor under the control of a growth-sensitive promoter. In this way inhibition of the process gene will decrease as growth rate raises (as sRNA expression will decrease) or as process output falls due to reduced activation of the sRNA expression. Here the sRNA promoter receives both ‘process-based’ and ‘growth-based’ signals (see Fig. S26b for controller topology). We name this new multi-input controller $CLp_A TL\lambda PF$ and subjected it to the same analysis as all other controllers in the manuscript. We find that this proposed controller topology enhances circuit performance compared to open loop but not compared to other controllers already present in the manuscript. We have described these new results in the main text alongside the other multi-input controllers. We first define the new controllers (including one developed in response to the first reviewer’s comment 5):

“(v) $CLp_A TL\lambda PF$ (PF := protein-free) uses both protein-based feedback and growth-based feedback to co-operatively prevent translation of the process gene by putting the sRNA gene directly on a growth-sensitive promoter (Fig. S26b).”

Outputs, parameters and robustness plots are given for both controllers in Fig. S26, S27 and S29, analogously to other controllers considered throughout the paper. We have amended the main text with the following analysis of these two new controllers:

“Although $CLp_A TX\lambda TX$ enhances performance versus open-loop, it performs worse than systems

(i)-(iii) without providing a notable improvement in robustness: changes in P_0 , $\tau_{\pm 10}$ and τ_{50} have the standard deviations 11.1% (28.1%), 19.1% (28.6%) and 4.2% (10.1%) (Fig. S26, navy). 80.6% of controllers retain $\tau_{\pm 10} = \tau_{90}$. At the same initial output P_0 , this controller is more highly expressed (larger ω_A) with weaker control in the intra-circuit component and stronger control in the growth-based component (larger k_λ , larger k_{B1}) (Fig. S27, navy). $CLp_A TL\lambda PF$ similarly enhances performance versus open-loop, but not to the extent of systems (i)-(iii). However, it does offer an improvement in robustness, with changes in P_0 , $\tau_{\pm 10}$ and τ_{50} having the standard deviations 6.6% (16.7%), 3.4% (8.6%) and 3.4% (8.6%), with 100% of designs retaining $\tau_{\pm 10} = \tau_{90}$ (Fig. S26, lime). However, this system requires a very high growth-based control strength (large k_λ) that may be difficult to achieve in practice (Fig. S27, lime). A comprehensive analysis of the robustness of all multi-input controllers is described in Supplementary Note 3 (Fig. S28-S29, Table S1).”

Discussion

9. Page 22, line 2: “without penalising output P_0 ” is not exactly true, since the negative feedback necessarily reduces the output level, but is true for a given set point level compared to OL equivalents.

We have clarified this by replacing “without penalising output P_0 ” with “compared with an open-loop system of equivalent output P_0 ”.

10. Can the authors comment on scalability of the controller designs for systems with increasing numbers of genes to control, and for situations where dynamic gene regulation may be involved?

We anticipate the majority of our conclusions will hold for complex systems. For example, we anticipate that negative feedback which utilises a circuit/network node output will extend the short-term lifespan of the node while one which utilises growth as an input will extend the node’s medium-term life. We also anticipate that our observation that translational, or sRNA-based, systems show enhanced performance over transcriptional, or TF-based, systems due to the reduced burden of the controller. The extension of our approach to more complex systems is an area we are actively exploring. We have amended the discussion of our manuscript as follows:

“Here, we have conducted a comprehensive analysis of feedback controllers when applied to a process consisting of a single gene over repeated batch cultures. This approach replicates previous experimental investigations of synthetic gene circuit evolution [4, 10, 20] and, to some extent, mimics bioproduction processes for single products. Controlling the evolutionary stability of more complex gene circuits represents a more significant challenge as the dynamics of our

negative feedback controllers and the complex processes may interact. It is common practice in control engineering to design a bespoke controller for each individual process. Our controllers can be scaled to stabilise gene expression in systems where gene expression levels are desired to be approximately binary (such as logic gates, activation cascades or induction systems), provided there are sufficient experimental choices of inhibitory regulator (e.g. orthogonal transcription factor proteins or sRNAs) - albeit with controller kinetics adjusted to suit the new process. Whilst controller designs will need to be optimised for the new processes, we expect our general design rules to hold: that negative feedback will stabilise expression over evolutionary time scales, that translation-based systems will offer superior performance over transcription-based systems (and that the latter may in fact reduce evolutionary stability) and that systems which seek to control function at the population level will be less effective. The choice of intra-circuit or growth-based feedback (and associated differences in design robustness) remains an area for future work. Engineering controllers to stabilise the evolutionary stability of circuits with time-varying dynamics (such as oscillators) represents an additional challenge given that the introduction of the new controller dynamics may abolish the desired circuit behaviour [41]. Engineering control systems to make such systems more robust and reduce the interactions between host and circuit remains an open question.”

Figures

11. Fig 2c-e: Include a descriptor in the legend about what the different colors mean. They can be inferred from Fig. 1 but this will improve clarity.

We have added a new panel Fig. 2b which serves as a legend and clearly demonstrates the colours.

12. Fig 3c-d: Include descriptor of what “x” means in the plots – I’m assuming it corresponds to the single timecourse in panel (b).

The reviewer is correct. We have refined the caption for panels (b-d) as follows:

“(b) Time-series of population-wide output P over time for an open-loop system (black, dashed, $w_A = 4.0 \text{ mc min}^{-1}$) and representative control systems of equivalent initial output P_0 . (Intra-circuit: red, $w_A = 10^3 \text{ mc min}^{-1}$, $k_A = 4.5 \times 10^2 \text{ mc}$. Growth-based: orange, $w_A = 87 \text{ mc min}^{-1}$, $k_\lambda = 6.3 \times 10^{-2} \text{ min}^{-1}$. Population-based: lilac, $w_A = 16 \text{ mc min}^{-1}$, $k_P = 5.2 \times 10^8 \text{ mc}$). (c-d) A large number of designs were generated by varying the maximal transcription rate w_A and control parameters k_u . Against the initial output P_0 , we plot the percentage change in (c) $\tau_{\pm 10}$ and (d) τ_{50} vs an open-loop system of equal initial output. Points marked with an X

correspond to the time-series plots in (b). These were selected as points on the Pareto fronts simultaneously optimising P_0 , τ_{50} and $\tau_{\pm 10}$, with initial output P_0 closest to 2×10^9 molecules.”

13. Fig 3e-f, S20e-f: Shading the colored and outlined boxes differently for each mutational state would help to connect the information in (e) to that in (f). Another option would be combining into a single plot of p_A vs λ (in which case the T0 vs T50 boxes would be different shades)

We have added colour gradients with a colour bar legend in Fig. 3e,f and Fig. S20e,f to clearly show which states are more/less functional. The caption for panels (e-f) has been updated as follows:

“(e-f) For circuits corresponding to trajectories in (b), (e) maximum protein production per cell p_A and (f) maximum growth rates λ across the first day (solid) and day where τ_{50} is reached (grey outline) for each mutation state. Lighter squares represent less functional strains. The horizontal line represents a non-functional strain.”

14. Fig 4f-g shows reduced T10 and T50 at the highest levels of P0 – does this imply that the controllers are decreasing robustness of output levels in that regime?

Fig. R1 here presents the time series output P of one of these controllers with very high initial output P_0 and worse performance than open-loop. For this open-loop system where initial output P_0 is at its maximum, burden is as high as the cell can withstand (i.e. this is the point beyond which increasing expression harms output). To achieve a comparable level of output, the closed-loop system must have a very low binding rate b_B so that there are very few controller proteins p_B (Fig. S7). The internal dynamics of the open-loop and closed-loop systems are therefore very similar, with equal initial output P_0 but a marginally higher burden in the closed-loop system due to the additional production of p_B . This increased burden means mutating away from the fully functional state provides a greater growth advantage, outweighing the benefit of control and leading to a reduction in $\tau_{\pm 10}$.

15. Fig 4i-j, 6h-i, 8j-o, S12, S15, S18: I think it would be more clear if the shading of +/-10% was dark grey (for uniformity) or a darker shade of the variant color itself, rather than using the dark red shade associated w/ CLpATL

We have replaced the red shading in these panels with a dark grey. Note also that these panels have now been moved to the supplementary material as Figs. S5, S8, S19, S28, S29, as those in the main text have been replaced with standard deviation plots in accordance with other comments.

16. Fig 4i-l: not explicitly referenced in the main text (presumably should be on Page 11, paragraph 2).

We have reviewed the manuscript to ensure all figures are referred to. As outlined in the response to other comments by both reviewers, the original Figs. 4i-l have been replaced with new standard

Figure R1: Time-series total output P over time for a closed-loop CLp_{ATL} controller (red) with very high initial output P_0 which performs worse than an open-loop system of equivalent output (green).

deviation plots as Figs. 4g-l. These are now explicitly referenced in the main text during the robustness analysis, e.g. the CLp_{ATX} case on page 11:

“Here, optimal controllers are very robust to parametric variation, with changes in P_0 , $\tau_{\pm 10}$ and τ_{50} having standard deviations of 5.6% (13.9%), 3.3% (6.8%) and 2.9% (7.0%) respectively compared with the original optimal systems when parameters were varied by $\pm 10\%$ ($\pm 25\%$) (Fig. 4g-i).”

Equivalent figures later in the paper are now referenced in the same way.

17. It’s also relatively unclear how to interpret beyond what is said in the main text, since the graphs do not directly show SD/CV, but rather the raw values. The graphs here and comparable ones elsewhere (e.g. Fig 6hi, Fig 8j-o, S12, S15, S18) could be improved by adding an inset measurement of SD/CV. In light of this comment and that of Reviewer 1 (comment 4), we have significantly improved our robustness analysis throughout the paper and overhauled all of the corresponding figures. This is detailed in our response to that comment.
18. If space/format permits, consider moving Supp Table 1 to main text – it’s very helpful as a summary of this robustness point

Based on the comments of both reviewers, we have rewritten our robustness analysis to improve its presentation by introducing new figures and a comprehensive analysis in Supplementary Note SN3. As we now show the standard deviation values present in the Table S1 as figures in the main text, we have chosen to leave Table S1 in the supplementary material. We have expanded Table S1 to include all robustness analysis results for all controllers and refer to it in new Supplementary Note 3.

19. Fig 5: Indicate somewhere that $mc := \text{molecules}$, for clarity

We have included this explicitly just after the definition of P on page 5: “Variables are given in molecules per cell (mc/cell)”. We have also been more clear throughout the paper to explicitly note when variables are being given on a population scale in molecules or mc (e.g. P) or on a per-cell basis in molecules per cell or mc/cell (e.g. p_A).

Supplement

20. Equation 28 (and other equivalents w/ multiple proteins from one mRNA, like eq 34): As-written, it appears that the translation of one ORF (i.e. proteins A vs B) affects the translation of the other due to mutually-exclusive formations of complexes w/ ribosome (CA, CB). This might not significantly affect the model, but translation of each ORF should be decoupled, as the ribosome can bind to each RBS independently.

The reviewer is correct; we have neglected co-translation of both ORFs and assume that each mRNA is bound by a single ribosomes. This assumption simplifies our model and has been made by many established coarse-grained host modelling frameworks. To clarify the assumption we have modified the text related to Equation 28 as follows:

“In line with the host model and other established models which do not account for polysome formation, we assume that each transcript mA can be bound by a single ribosome at either of two ribosome binding sites to form the corresponding complex cA or cB [31,19,34,44].”

To verify that our assumption does not significantly affect our results, we have carried out a re-analysis of the $CLp_A TX$ system using a new model where ribosomes can co-translate circuit/process and controller proteins. The modified model is shown below. Briefly, we consider a single promoter which gives rise to an mRNA which can be bound at at the process RBS (position ‘A’) and controller RBS (position ‘B’). To model this process we consider the production of two mRNAs m_A , corresponding to the process, and m_B , corresponding to the controller. The birth rates of m_A and m_B are the same ($T_{X_A(e)}\Theta(p_B)$) and therefore are functionally the same mRNA molecule within the model - here one can consider the m_X species to be an RBS and codon sequence of the mRNA rather than the mRNA itself. Therefore

when a ribosome binds with the transcript at one binding site, it doesn't prevent another ribosome binding at the other ribosome binding site. The complete model of the system is as follows:

$$\dot{m}_A = T_{X_A}(e)\Theta(p_B) + T_{L_A}(c_A, e) - b_A R m_A + u_A c_A - (\lambda + \delta_{m_A}) m_A, \quad (1)$$

$$\dot{c}_A = -T_{L_A}(c_A, e) + b_A R m_A - u_A c_A - \lambda c_A, \quad (2)$$

$$\dot{p}_A = T_{L_A}(c_A, e) - \lambda p_A, \quad (3)$$

$$\dot{m}_B = T_{X_A}(e)\Theta(p_B) + T_{L_B}(c_B, e) - b_B R m_B + u_B c_B - (\lambda + \delta_{m_B}) m_B, \quad (4)$$

$$\dot{c}_B = -T_{L_B}(c_B, e) + b_B R m_B - u_B c_B - \lambda c_B, \quad (5)$$

$$\dot{p}_B = T_{L_B}(c_B, e) - \lambda p_B. \quad (6)$$

We parametrise the system identically to the original *CLp_ATX* system and perform a multi-objective optimisation in the same way on the parameters ω_A , b_B and k_B (Table S8). Comparing our original results with the results from this new model shows very similar performance for $\tau_{\pm 10}$ and τ_{50} (Fig. R2a,b). Although we see minor changes in the controller design rules with slightly smaller values of ω_A and b_B required to achieve the same initial output P_0 (Fig. R2c-e) due to the lack of competitive exclusion, we do not see any qualitative changes in our previously established design rules. These results give us confidence that the lack of co-translation/polysome formation in our framework has not significantly driven our observations.

21. Equation 34: Should the (pB) be there in the TXA term?

We have corrected this typographical error.

22. Here and in several other eqs (including 39, 50, 56, and more), it appears that the species rC should be mC?

We thank the reviewer for highlighting these typographical errors. The paper has now been reviewed and all typographical errors have been corrected. This specific error originated as we initially denoted the sRNA variable as m_C but we thought this may imply it was an mRNA and so it was renamed to r_C . There are not mRNAs of 'gene C' within our models.

Reviewer 2 (Remarks on code availability):

README file is included and clear, code is clear on quick examination. I did not run the code to test it, though (do not currently have MATLAB license).

We thank the reviewer for their comments on our provided models.

Figure R2: Repeating the optimisation of $CLp_A TX$ system with a model which allows for ribosomes to simultaneously bind at both ribosome binding sites via the additional variable m_B . (a-b) Optimal outputs for the original model (blue) and new model (red). Initial output P_0 is plotted against (a) $\tau_{\pm 10}$ and (b) τ_{50} . (c-e) Optimal parameters: (c) ω_A , (d) b_B , (e) k_B .

Response to reviews for
Byrom and Darlington “Genetic controllers for enhancing the evolutionary longevity of synthetic gene circuits in bacteria”

We thank the reviewers for their time evaluating our manuscript and for their final comments on our paper. We thank the editorial office for their suggested changes to the text and figures. This letter details the changes requested.

REVIEWERS' COMMENTS

Reviewer #1 (Remarks to the Author):

The authors have fully addressed all my comments.

We are pleased to have addressed the reviewers comments in full.

Reviewer #2 (Remarks to the Author):

Thank you to the authors for your revisions and consideration of all reviewer comments. As with the rest of the manuscript, the responses and revisions were very detailed, well-considered, and high-quality. I would recommend publication, with very minor typographic revision of the following:

Fig S26 is referenced in main text prior to S24-5, so consider re-arranging these figures in the SI

We have reordered the figures in the Supplementary Material to reflect the order they are referenced in the main text.

Refs for most supplementary tables in main text are missing

We have moved the Supplementary Tables from Supplementary Note 1 so that they appear before the Supplementary Notes. There have been minor changes to the main text to refer to the Supplementary Tables.

Reviewer #2 (Remarks on code availability):

No additional comments.